computer modelling and simulation/civil engineering/geology

ultrasonic vibration, granite, fatigue behaviour, PFC2D, crack evolution, fatigue damage model

**Authors for correspondence:**
Qiongqiong Tang
e-mail: tangqq17@mails.jlu.edu.cn
Meiyan Wang
e-mail: 9101037@qq.com

# Experimental and numerical investigation of the fatigue behaviour and crack evolution mechanism of granite under ultra-high-frequency loading

Yu Zhou[1], Dajun Zhao[1], Qiongqiong Tang[1] and Meiyan Wang[1,2]

[1]Complex Condition Drilling Experiment Center, Jilin University, 130012 Changchun, People's Republic of China
[2]Shandong Vocational College of Science and Technology, 262700 Weifang, People's Republic of China

YZ, 0000-0001-8436-8766; QT, 0000-0001-6518-7767;
MW, 0000-0001-9415-6054

Assisted ultrasonic vibration technology has received great interest in the past few years for petroleum and mining engineering related to hard rock breaking. Understanding the fatigue behaviour and damage characteristics of rock subject to ultra-high-frequency loading is vital for its application. In this research, we conducted ultrasonic vibration breaking rock experiments combined with an ultra-dynamic data receiver and strain gauges to monitor the development of strain in real time. The experimental results show that the strain curve is U-shaped, and it can be divided into three stages: the strain first decreases, then remains steady (with light fluctuations) and finally increases. The sample first underwent compressive deformation, and no rupture occurred. As the vibration continued, the compressive deformation decreased with the initiation and propagation of cracks, and fragmentation occurred. To elucidate the crack evolution mechanism of the granite specimens, numerical simulations were performed using particle flow code in two dimensions (PFC2D), and an improved fatigue damage model based on the flat-joint contact model was proposed. The numerical results indicate that this model can effectively reproduce the fatigue characteristics of hard brittle rocks under ultrasonic vibration. By analysing the stress and strain fields and cracking process, the crack evolution mechanism in the brittle hard rock under ultra-high-frequency loading is revealed. These experimental and numerical results are expected to improve the understanding of the fragmentation mechanism of rock under assisted ultrasonic vibration.

# 1. Introduction

Due to the fact that the hard and brittle rocks with high strength are widely encountered in underground projects such as tunnelling, petroleum and geological drilling (e.g. granite), rapid breaking of hard rocks is a widespread issue that needs to be optimized [1–3]. To quickly break hard rock, technological advancements have used the assisted ultrasonic vibration technique [4] and Wiercigroch *et al.* concluded that an introduction of ultrasonic vibration significantly enhances drilling rates compared to the traditional rotary type method. The reason for this phenomenon is that rock produces significant fatigue damage under ultrasonic vibration (ultrasonic cyclic loading).

At present, the fatigue and mechanical behaviour of rock under cyclic loading has been investigated deeply. Wang *et al.* [5] found that compared with the loading cycle number, the axial strain could be a better option to describe the fatigue behaviour of rock. In the low-stress region, Hsieh *et al.* [6] concluded that the stiffness of the rock will first increase due to the crack closure and then decrease when crack growth occurs after repeated loading. Yang *et al.* [7] used the P- and S-wave velocities based on the ultrasonic wave velocity method to reflect the sandstone behaviour (i.e. elastic modulus and the shear modulus) under cyclic loading condition and found that rock specimens stiffen during loading and soften during unloading. Jiang *et al.* [8] found that the porosity and pore size of marble rock gradually decrease under cyclic loading, which indicated that its framework becomes compacted. Apart from marble rock, red sandstone was also investigated by this research group, and the authors concluded that the damage variables showed a strong nonlinear relationship and increased with the loading cycles number [9]. The energy evolution of rock under cyclic loading has been investigated by many researchers [10–14], who found that the total absorbed energy mainly converts into elastic energy. The rest stored energy is released in the form of dissipated hysteresis energy for the propagation and coalescence of microcracks.

The above-mentioned studies of rock fatigue behaviour were all under low-frequency cyclic loading. However, ultrasonic vibration is characterized by its ultra-high frequency (greater than or equal to 20 kHz), which is of the same order of magnitude as the natural frequency of hard rock [15]. If the excitation frequency is the same as the natural frequency of rock, rock will be resonant. Besides the high-frequency characteristics, low amplitude (10–30 µm) and fast energy loss are the other two [16], and these characteristics are quite different from the traditional loading conditions. Thus, it is significant and essential to further study the fatigue behaviour and crack evolution mechanism of rock under ultrasonic vibration. However, nowadays, there is still a lack of research on this subject. Yin *et al.* [17] conducted the uniaxial compressive strength (UCS) experiments to obtain the development of granite rock strength after ultrasonic vibration; the results demonstrated that this technique can effectively reduce the strength of hard granite when the static loading exceeded the static pressure threshold as the fatigue damage occurred. Zhao *et al.* [18] divided the granite rock sample body into three zones when subjected to ultrasonic vibration: the fracture zone, plastic deformation zone and elastic deformation zone. Zhou *et al.* [19] investigated the effect of static loading on the natural frequency of the granite rock sample when subjected to ultrasonic vibration and found that the inherent frequency of rock increased logarithmically with the magnitude of static loading. The current research status indicates that the research on the application of ultrasonic technology in hard rock breaking is still in the first stage, the fatigue behaviour and crack evolution mechanism of rock under ultrasonic vibration are still unclear and, therefore, it is quite necessary to undertake research on these subjects.

Clearly, the evolution of microcracks plays an essential role in the rock failure process, and it is necessary to investigate the cracks' behaviour during the fatigue process. With the rapid development of computer technology and advanced numerical techniques, more details, especially the characterization of crack growth, can be determined. For example, Hu *et al.* [20] used LS-DYNA to study the rock fragmentation mechanisms when cut by a cutter and concluded that there are three fracture modes of rock breaking: forward slip, no slip and backward slip. Yang *et al.* [21] investigated the rock-breaking mechanism under supercritical (SC)-$CO_2$ jet impacting based on the smoothed particle hydrodynamic finite-element method (SPH-FEM). Numerical results showed that there were two failure forms for rock under SC-$CO_2$ jet impacting presented: erosion and exfoliation. Wang *et al.* [22] studied the influence of the intermediate principal stress on the mechanics of three-dimensional crack growth from a spherical pore based on the ABAQUS, and the results show that with the intermediate principal stress above the threshold value, the directions of most of the secondary principal stresses (tensile) along the lateral surface of the initial pore are roughly perpendicular to the

intermediate principal compressive stress direction. Yang *et al*. [23] investigated the influence of cracks on the mechanical properties of rock using Franc3D and found that the strength decreases with a decrease in the crack dip angle and an increase in the crack density. Rock failure process analysis was used to study the internal damage evolution of rock under uniaxial compression and tension [24]. Recently, the discrete element method (DEM) has received considerable attention for its advantages in simulating the microcrack evolution and discontinuity property of rock materials. Song *et al*. [25] determined the failure position of a tunnel based on a DEM simulation and concluded that it is an efficient method to predict the instability in the rock surrounding a tunnel with a complex geological condition. Based on a novel clustered assembly approach, Li *et al*. [26] studied the rock fragmentation mechanism and failure process induced by a wedge indenter and found that the initial void and defects existing in the model exert a significant effect on the coalescence and propagation of a median crack as well as on the stress concentration. Based on the granite rock model established by PFC2D, Tian *et al*. [27] verified numerically that shear cracks are more difficult to initiate in coarse granite specimens than in fine granite specimens because there are less grain boundary contacts in coarse-grained materials. To enhance the understanding of failure mechanism of rock, Huang *et al*. [28] carried out conventional triaxial compression tests simulation by using particle flow code in three dimensions (PFC3D), and the results showed that different lengths of the sample result in different failure patterns.

In this study, one aim is to investigate the fatigue behaviour of granite under ultrasonic vibration based on the strain test using an ultra-dynamic data receiver and strain gauges in real time. Another aim is to use a two-dimensional discrete numerical code, PFC2D, to simulate the mesoscopic laboratory test and explore the crack evolution law in the process of ultrasonic vibration breaking rock (UVBR), which is difficult to observe in dynamic experiments. The long-term aim is to use the established model to examine the loading parameters (i.e. frequency and amplitude of a dynamic load and the magnitude of static loading) on the fragmentation efficiency and to know more about rock-breaking mechanisms. This paper is organized as follows. Section 2 introduces the experimental methodology, including the loading method and facility and the specimen used. Section 3 reports the experimental results of the ultrasonic vibration tests and analyses the fatigue behaviour of rock. In §4, details of the rock model generation and micro-parameter calibration procedure are presented. Section 5 proposes the numerical methodology to model the UVBR process, and it concludes with four key points. The mechanical fatigue behaviour and crack evolution results of the rock specimen is analysed and revealed in the following section. Finally, §7 summarizes the conclusions drawn from this study.

# 2. Experimental methods

## 2.1. Sample preparation

The rock samples used in the experiments were processed from general fine-grained granite from Jilin Province, China, a common type of hard brittle rock. Its mineralogical composition was determined by X-ray diffraction analysis (figure 1), which shows that this rock contains quartz, albite, orthoclase and hydrobiotite. The samples were shaped into standard cylindrical blocks with a diameter of 36 mm and a height of 72 mm. The average grain size is approximately 1.1 mm, as shown in figure 2. The approximate proportions of the different minerals are feldspar (29.4%), quartz (50.6%), orthoclase and hydrobiotite (18.3%) and others (1.7%). According to the ISRM [29] standard regarding uniaxial compression tests in the laboratory, the slenderness ratio (the ratio of the diameter to the height) of the sample is suggested to be in the range of 2.0–2.5. Therefore, the sample prepared in our work obeys the ISRM standard. Simultaneously, to reduce the effect of the rock heterogeneity on the experimental results, samples were analysed using the knocking method to examine their natural frequencies. Samples with similar natural frequencies from 26 to 27 kHz were picked out, and one of their natural frequency test results is shown in figure 3.

## 2.2. Testing set-up

The test equipment comprises an ultrasonic vibration device and a data acquisition device, as shown in figure 4*a*. The loading apparatus is composed of an ultrasonic power source (1), an ultrasonic vibrator (2) and a static loading device (3). The loading method in this test is shown in figure 4*b*. An ultrasonic dynamic load combined with a vertical static load were applied to the samples. The static loading was applied with weights, and each of them weighs 10 kg. They were placed on the top of the bracket. The

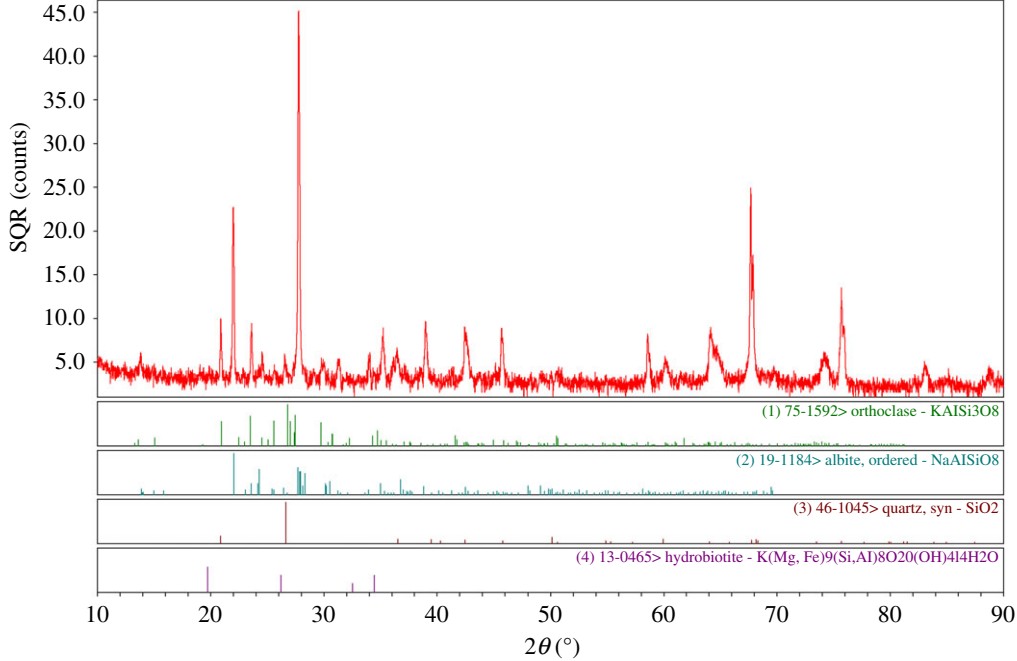

**Figure 1.** X-ray diffraction showing mineral compositions of granite.

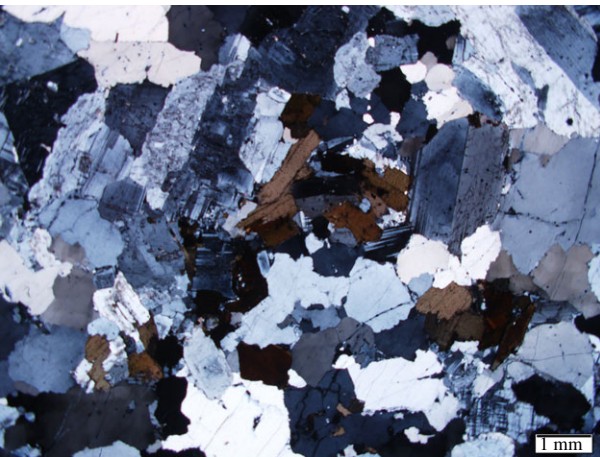

**Figure 2.** Polarized photomicrograph showing the grain size of the granite.

data collection apparatus includes a computer (5) and an ultra-dynamic data receiver (4) combined with a strain gauge (6), and it was used to collect axial strain data in real time. The strain gauge was placed on the top-side surface of the cylindrical rock specimen, where the damage was severest.

The static loading and ultrasonic vibration frequency were 200 N and 30 kHz, respectively, under which the crushing efficiency is higher based on the previous experiments [30]. The strain behaviour of the rock was monitored until fragmentation occurred. For this test, it contains six samples. All the experiments were conducted at room temperature.

# 3. Experimental results and discussion

Figure 5 shows the development of axial strain for six samples. Although the strain curves are not identical due to the heterogeneity of rock, it is easy to conclude the development law. The axial strain curve is U-shaped, and it can be divided into three stages: the strain first decreases, then remains steady (with light fluctuations) and finally increases. The negative strain in the figure represents compressive deformation, and the positive one represents tensile deformation.

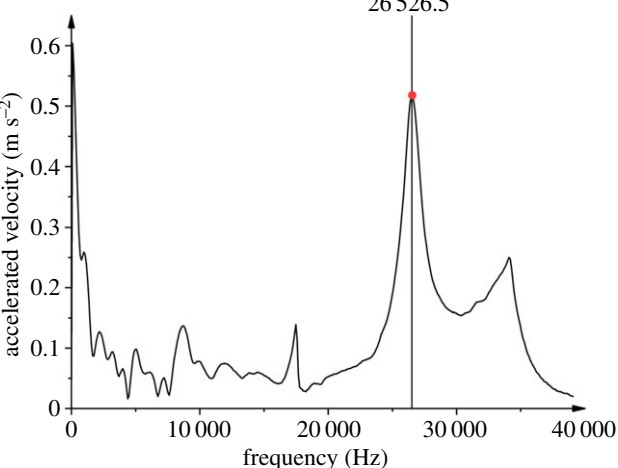

**Figure 3.** Natural frequency of the rock sample.

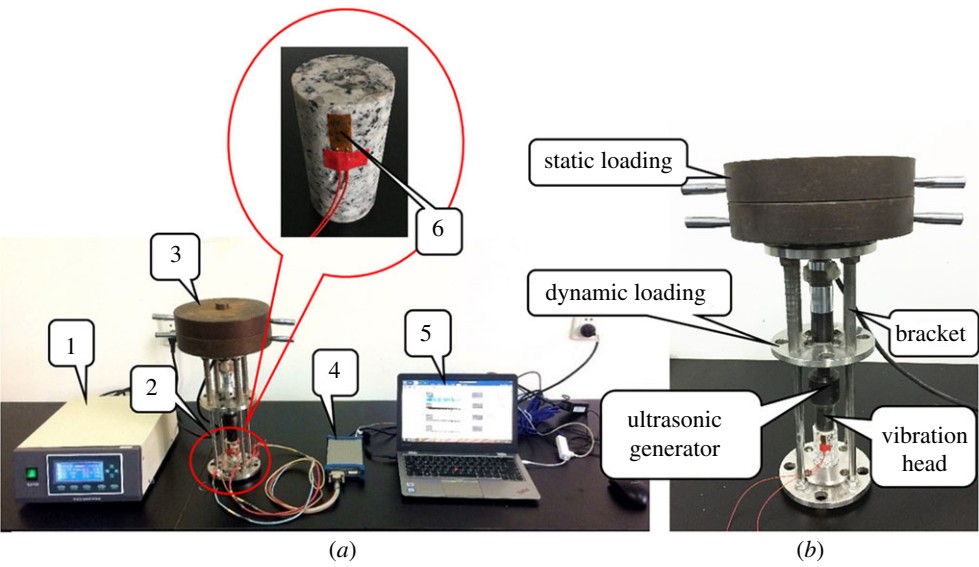

**Figure 4.** Ultrasonic vibration strain test: (*a*) ultrasonic vibration and strain testing equipment and (*b*) loading method.

In the first stage, as the surface of the open crack might not be smooth, the sharp points might break when subject to ultrasonic vibration [31]. The axial strain reduces because of the generation of breakage accompanied by the crack closure. Some energy is consumed in this process and irreversible compressive deformation gradually increases. Thus, the sample becomes more 'compacted' [32]. In other words, the modulus of rock would increase.

As the vibration continues, due to the highly concentrated stress, new cracks gradually form and the initial cracks would reopen and propagate. Thus, the slope of the strain curve decreases. These cracking behaviours would result in a decrease in rock modulus and cause rock expansion in the axial direction. In the last stage, as the cracks continue to propagate and coalesce, the magnitude of the expansion would become larger, and the strain increases.

As the strain gauges would fail when large fragments first occur, to observe the complete fragmentation process of the rock sample under ultrasonic vibration, we conducted the fragmentation monitoring test and it contains six samples. One general case is shown in figure 6. First, a relatively small fragment zone occurs at the edge of the sample. Then, the fragment zone continues to extend from the initial region to the adjacent area, and the fragmentation zone becomes larger. Finally, macroscopic damage distributes over the whole top edge and further develops into the centre. According to the fragmentation pattern, it can be concluded that there exists an effective region for this technique. Inside this effective region, there is obvious damage, while there is almost no damage in the other place.

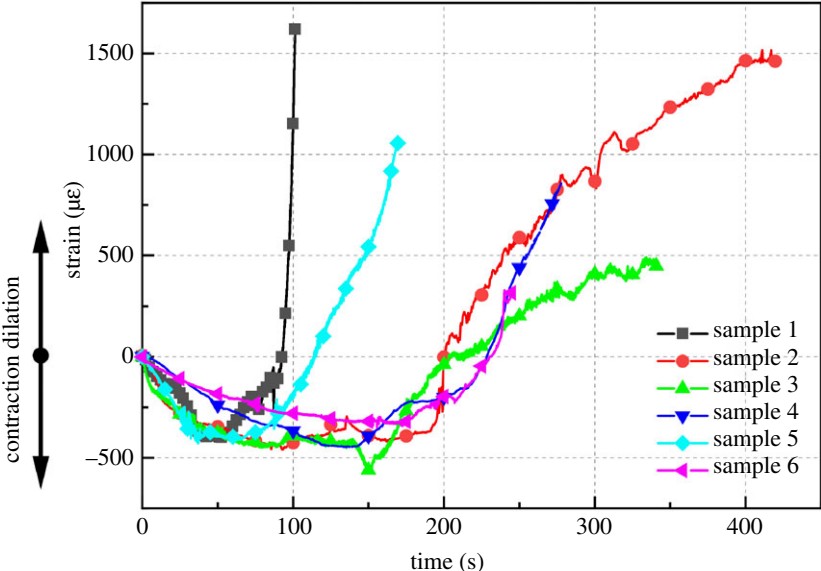

**Figure 5.** Strain data and failure pattern of six samples in vibration strain test (the negative strain represents the contraction of sample and the positive strain represents the dilation of sample).

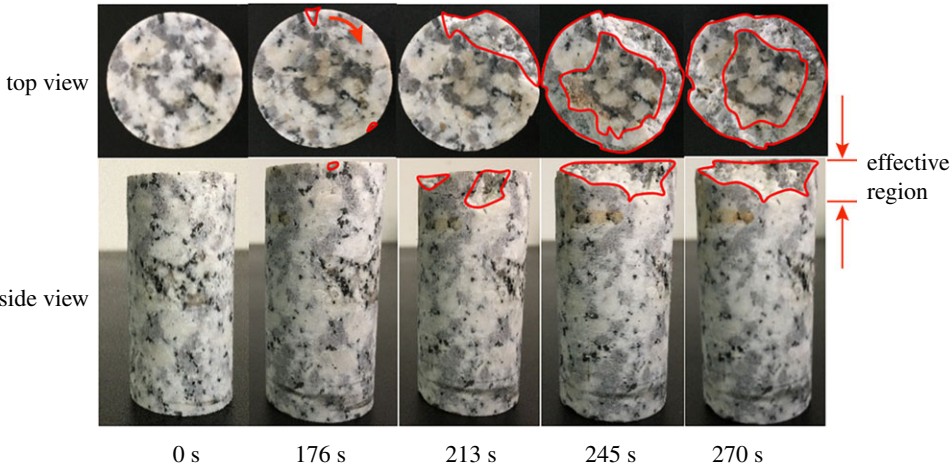

**Figure 6.** Fracture mode of the samples (area enclosed by red line represents the destruction zone and the arrow points in the destruction direction).

Figure 7 illustrates the development in the strain amplitude. The strain amplitude is the difference between the maximum and the minimum axial strains in each second. The strain amplitude was obtained and combined with the development of the elastic modulus to track the stress amplitude, and the stress amplitude can greatly affect the fatigue behaviour of rock sample [33]. Throughout the early vibration process, the strain amplitude increases, and as the vibration further continues, the strain amplitude reaches a maximum and then decreases.

In the early stage, due to the crack closure, the stiffness of rock increases further, causing the increase in the natural frequency. The change in natural frequency also affects the strain amplitude as the rock is reaching the resonance condition [34]; that is, the strain amplitude increases. Thereafter, during the early process of loading, as the sample compresses and is accompanied by crack closure, both the strain amplitude and stiffness of the rock gradually increase. Thus, the stress amplitude increases in this stage, and the fatigue process accelerates. After this early stage, closed cracks will reopen and propagate accompanied by the initiation of new cracks, which will lead to a decrease in the stiffness and natural frequency of rock, and further decrease the strain amplitude and stress amplitude. Thus,

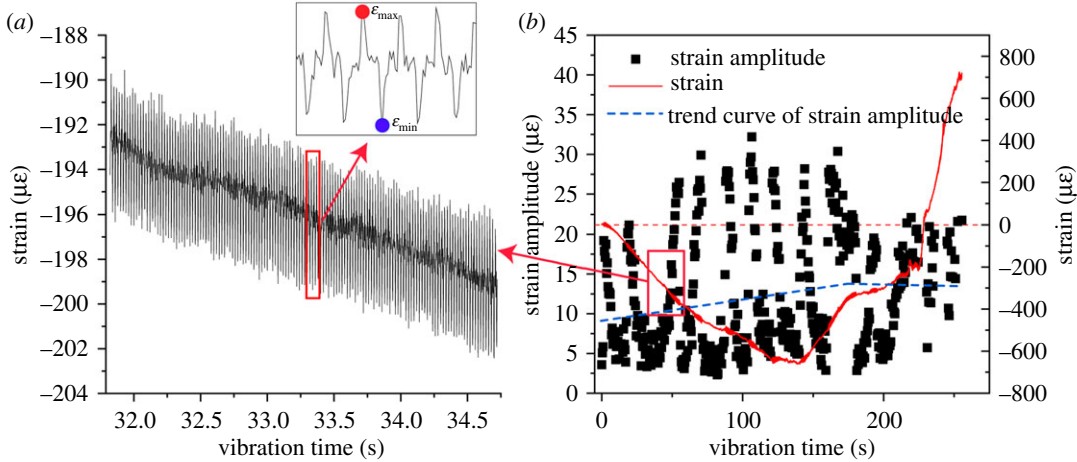

**Figure 7.** Change process of strain amplitude: (*a*) partial enlargement map of strain and (*b*) strain and strain amplitude curves.

to improve the fragmentation efficiency of this technology, the applied ultrasonic vibration frequency should be changed in real time to approach the natural frequency of the rock.

# 4. Particle flow modelling of granite

## 4.1. Flat-joint model

PFC, based on the principle of the DEM, can be used to study the evolution of cracks in brittle rocks [26]. Considering the computer performance, we used PFC2D in the present work. In PFC2D, a material is modelled as collections of particles (rigid circular discs). Each particle is in contact with the neighbouring particles via the contact model. The parallel-bond model has often been used to simulate crack initiation and propagation in rock or rock-like materials [26,35–37]. However, there are several issues concerning this contact model when simulating brittle hard rock [38]: (i) an unrealistically low unconfined compressive strength to tensile strength ratio, (ii) an excessively low internal friction angle, and (iii) a linear strength envelope. To solve these problems, Potyondy [39] proposed a new bond model called the flat-joint model (FJM). Wu, Xu and other researchers [38,40,41] have indicated that its high reliability for hard rock simulation based on UCS test and Brazilian tensile strength (BTS) test simulations. Thus, in this study, we chose the FJM to perform numerical simulations of granite rocks under ultrasonic vibration.

After a flat-joint contact is installed between two particles, forces are generated following the judgement of crack generation. The normal force is computed directly, and the shear force in an incremental fashion [42].

The normal stress $\sigma^{(e)}$ and shear stress $\tau^{(e)}$ acting on an element are given by

$$\left.\begin{array}{l} \sigma^{(e)} = \dfrac{-\overline{F_e^n}}{A^{(e)}} \\[2mm] \tau^{(e)} = \dfrac{\overline{F_e^s}}{A^{(e)}}, \end{array}\right\} \tag{4.1}$$

where $\overline{F_e^n}$ and $\overline{F_e^s}$ are the normal and shear forces acting on the element and $A^{(e)}$ is the area of the element.

The interface normal stress $\sigma_i$ is given by

$$\sigma_i(r) = \begin{cases} 0, & \text{unbonded and } g(r) \geq 0 \\ k_n g(r), & \text{otherwise} \end{cases}, \tag{4.2}$$

where $k_n$ is the normal stiffness and $g$ is the interface gap.

The normal force-update law for a flat-joint contact is as follows:

$$\bar{F}_e^n = \int_e \sigma \mathrm{d}A. \tag{4.3}$$

An analytical expression for this integral is used to update $\bar{F}_e^n$, which is then used to update $\sigma^{(e)}$ in equation (4.1).

The shear force-update law for a flat-joint contact is as follows:

$$\bar{F}_e^s = k_s A^{(e)} \delta_s, \tag{4.4}$$

where $k_s$ is the tangential stiffness and $\delta_s$ is the tangential displacement. Then, this result is used to update $\tau^{(e)}$ in equation (4.1).

The tensile strength is specified as $\sigma_b$. If the normal stress $\sigma^{(e)} > \sigma_b$, then the element breaks, and a tensile crack generates.

The procedure considering the shear strength based on the bond state is given below.

### 4.1.1. Unbonded

The shear strength obeys the Coulomb sliding criterion

$$\tau_r = \begin{cases} -\sigma^{(e)} \tan \phi_r, & \sigma^{(e)} < 0 \\ 0, & \sigma^{(e)} = 0 \end{cases}, \tag{4.5}$$

where $\tau_r$ is the residual friction strength and $\phi_r$ is the residual friction angle. If $\tau^{(e)} \leq \tau_r$, then the shear stress remains as $\tau^{(e)}$. Otherwise, the shear-strength limit is enforced, and slip occurs.

### 4.1.2. Bonded

The shear strength follows the Coulomb criterion with a tension cut-off

$$\tau_c = c_b - \sigma^{(e)} \tan \phi_b, \tag{4.6}$$

where $c_b$ is the bond cohesion and $\phi_b$ is the local friction angle. If $\tau^{(e)} \leq \tau_r$, the shear stress remains as $\tau^{(e)}$. Otherwise, the shear-strength limit is exceeded. The bond breaks, and a shear crack generates. Then, the residual friction takes effect.

## 4.2. Micro-parameter calibrations

The microscopic parameters calibrated from macro-properties obtained in the mechanical laboratory are considered to be an important procedure for PFC simulation. Only after this process, can the established model be used in the next stage.

### 4.2.1. FJM generation

To reproduce the granite sample, the 'clustered assembly approach' was used to generate a randomly distributed-grain granite sample. The model generation procedure is as follows:

Step 1: Two intact numerical specimens are generated first, and the scale of the numerical specimen is 36 mm in diameter and 72 mm in height in the uniaxial compression test simulation and 36 mm in diameter in the Brazilian test simulation. The particle size is controlled by predefined maximum and minimum radii ranging from 0.35 to 0.58 mm, considering the computer performance. Each numerical specimen is discretized into 3346 particles with 8513 contacts in the uniaxial compression test simulation and 1360 particles with 3763 contacts in the Brazilian test simulation.

Step 2: To reduce the complexity, it is assumed that there are three minerals in granite, and the contents of these three minerals (i.e. quartz, feldspar and biotite) are 31%, 51% and 18%, respectively. The clustered FJM in the UCS test simulation generates according to the particle extend method shown in figure 8. Each cluster in the model represents a rock grain. As the grain size decreases in the range of albite, quartz and biotite, in the present model, the particle number in each mineral cluster is assumed as follows: feldspar ($30 \pm 10$), quartz ($19 \pm 8$) and biotite ($12 \pm 5$). The first cluster starts to generate from two connected particles (figure 8b). It further extends until reaching the required size (figure 8c). Then, the second cluster starts to generate in the same way as the first one (figure 8c). Through repetition, the generation process stops until the sample is filled with clusters (figure 8d). At last, those clusters are grouped according to the mineral component proportions (figure 8d), and their density property is assigned according to their actual property (quartz: 2650, feldspar: 2600 and biotite: 2900 kg m$^{-3}$).

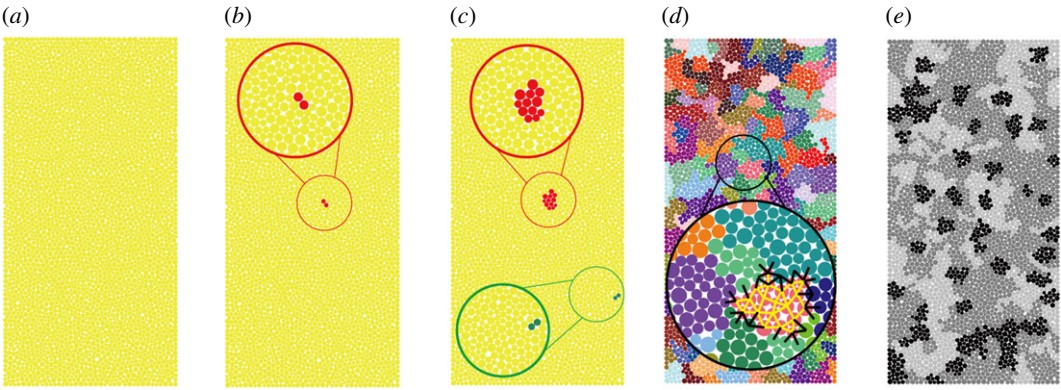

**Figure 8.** Particles extend cluster method: (a) the initial intact sample, (b) the beginning of the first cluster (red circles), (c) the first cluster finishes and the second one starts to generate (green circle), (d) the sample with all clusters (the black lines represent intergranular contacts and the yellow lines represent intragranular contacts) and (e) the distribution of minerals (light grey cluster represents quartz, dark grey feldspar and black biotite).

### 4.2.2. Calibration procedure

After establishing the clustered FJM, we need to import the contact information (i.e. micro-parameters) into the model. There are many micro-properties of the flat-joint contact (e.g. effective moduli, normal/shear stiffness, flat-joint tensile strength and cohesion) that need to be determined from macro-properties (e.g. Young's modulus, Poisson's ratio, UCS and BTS). Generally, for granite rock, Young's modulus decreases in the order of quartz, feldspar and biotite, and Poisson's ratio decreases in the order of biotite, feldspar and quartz. Similarly, the UCS decreases in the order of quartz, feldspar and biotite, and the BTS decreases in the order of feldspar, quartz and biotite [43,44].

Apart from the above-mentioned characteristics, the collapsing strength for the granular interior of granite is higher than the intergranular strength [37]. Thus, those contacts inside the clusters (intragranular contacts) are assigned a higher bonding strength ($\sigma_b = 18 \pm 1.8$ and $C_b = 100 \pm 10$ for quartz, $\sigma_b = 27.5 \pm 2.75$ and $C_b = 88 \pm 8.8$ for feldspar, and $\sigma_b = 12.5 \pm 1.25$ and $C_b = 60 \pm 6$ for biotite), while the strength of intergranular contacts, which link different clusters (figure 8d), is smaller ($\sigma_b = 9.5 \pm 0.95$, $C_b = 40 \pm 4$).

The uniaxial compression and Brazilian tests and corresponding simulations were performed to calibrate the micro-properties. When uniaxial compression test simulations were performed, the loading velocity of the upper and bottom walls is 0.05 m s$^{-1}$, and for the Brazilian test simulations, the loading velocity is 0.0075 m s$^{-1}$, under which the specimen could be regarded in a quasi-static state [40,45]. The loading wall width in the Brazilian test simulations is equal to two average particle diameters (2.68 mm), and to ensure the smoothness of the specimens, a circular wall with a resolution equal to 0.08 was used to generate samples, as suggested by Xu *et al*. [40]. The detail of the calibration procedure can be seen in [46].

The micro-parameters obtained after the calibration are listed in table 1. Furthermore, the calibrated simulated failure modes and macro-properties show excellent agreement with the laboratory test (table 2 and figure 9). Hence, the further simulation was based on this rock model.

### 4.3. Crack mechanisms in three mechanical tests

It is widely accepted that tensile cracking is the main damage mechanism of brittle rock in compression tests, such as the Brazilian test and UCS test [47]. In the UCS test, most of the tensile cracks formed are caused by compression. The tip of the pre-existing cracks endures tension under compression and induces tensile wing cracks. However, the samples in Brazilian tests locally generate tension force and hence it will split apart. For the direct tension test, tensile cracks are induced by direct tension from stretching the ends of the sample [48].

In this section, the orientation of cracks (the angle between the crack dip and horizontal direction) in the UCS, BTS and direct tension test simulations are obtained. The numerical samples in the tension test simulation are the same as those in the UCS simulation. The loading rate is set as 0.01 m s$^{-1}$. Figure 10 shows the tensile crack orientations from these three tests simulations. In the UCS and BTS tests, although the number of cracks varies greatly, the direction of most tensile cracks is in the range of 60–120° (the

**Table 1.** Micro-properties of balls and contacts.

| | quartz | feldspar | biotite |
|---|---|---|---|
| **general** | | | |
| volume composite, $V_{ratio}$ | 30.7% | 50.6% | 18.7% |
| **particle** | | | |
| minimum ball radius, $r_{min}$ (mm) | 0.35 | | |
| maximum/minimum, $r_{max}/r_{min}$ | 1.66 | | |
| ball density (kg m$^{-3}$) | 2650 | 2600 | 2900 |
| **intragranular contacts** | | | |
| bonded element fraction, $\varphi_B$ | 1 | | |
| slit element fraction, $\varphi_T$ | 0 | | |
| total number of elements, $N_r$ | 2 | | |
| stiffness ratio | 2.4 | 2.6 | 3.0 |
| effective moduli (GPa) | 78 | 50 | 30 |
| mean and s.d. bond tensile strength, $\sigma_b$ (MPa) | 18 ± 1.8 | 27.5 ± 2.75 | 12.5 ± 1.25 |
| mean and s.d. bond cohesion, $C_b$ (MPa) | 100 ± 10 | 88 ± 8.8 | 60 ± 6 |
| local friction angle, $\phi_b$ (°) | 3 | 5 | 8 |
| residual friction angle, $\phi_r$ (°) | 7 | 10 | 13 |
| **intergranular contacts** | | | |
| stiffness ratio | 3.1 | | |
| effective moduli (GPa) | 23 | | |
| mean and s.d. bond tensile strength, $\sigma_b$ (MPa) | 9.5 ± 0.95 | | |
| mean and standard deviation bond cohesion, $C_b$ (MPa) | 40 ± 4 | | |
| local friction angle, $\phi_b$ (°) | 10 | | |
| residual friction angle, $\phi_r$ (°) | 15 | | |

**Table 2.** Granite properties from experiments and simulations. The $m$ denotes the number of samples in the UCS and Brazilian tests; the $n$ denotes the number of samples with different bond strength distributions in the simulations.

| property | experiment results | simulation results |
|---|---|---|
| Young's modulus $E$ (GPa) | 52.3 ± 12.2 ($m = 10$) | 52.7 ± 0.26 ($n = 10$) |
| Poisson's ratio $m$ | 0.26 ± 0.05 ($m = 10$) | 0.255 ± 0.006 ($n = 10$) |
| Uniaxial compressive strength (UCS) (MPa) | 116 ± 19 ($m = 10$) | 116.3 ± 3.1 ($n = 10$) |
| Brazilian tensile strength (BTS) (MPa) | 8.1 ± 1.5 ($m = 10$) | 8.1 ± 0.17 ($n = 10$) |

direction here is the direction of crack propagation, figure 9a), which are parallel and subparallel to the loading direction. In the direct tension test, the crack orientation law is different from that in the other two tests, in which the mean orientations of the tensile cracks are in the ranges of 0°–30° and 150°–180°, which are perpendicular and subperpendicular to the loading direction. The rose diagram of crack orientations shows that the crack orientation obtained by the PFC simulations can effectively reflect the mechanisms of crack evolution.

# 5. DEM modelling of the UVBR process

In this section, the procedure to simulate UVBR is proposed. To ensure the pre-requisites determining the success of modelling, four key points should be considered.

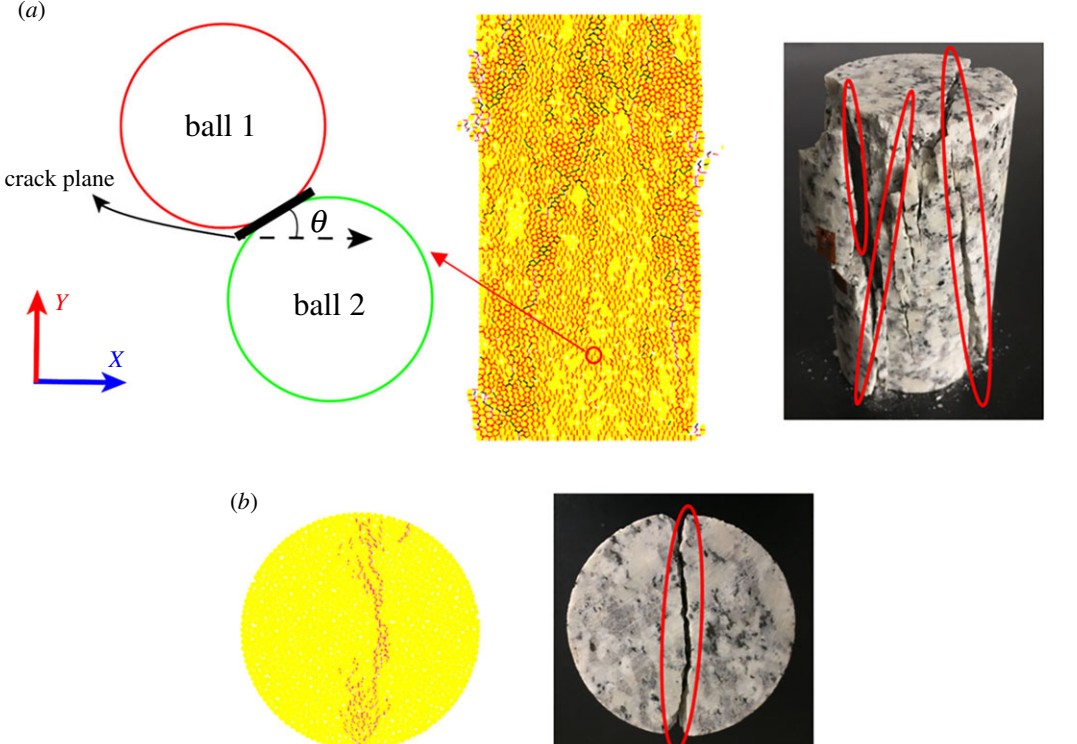

**Figure 9.** Comparison of failure modes obtained by laboratory tests and numerical simulations: (*a*) uniaxial compression test and (*b*) Brazilian test (the black line represents shear crack, while the red line is tensile crack; the yellow background is the rock sample).

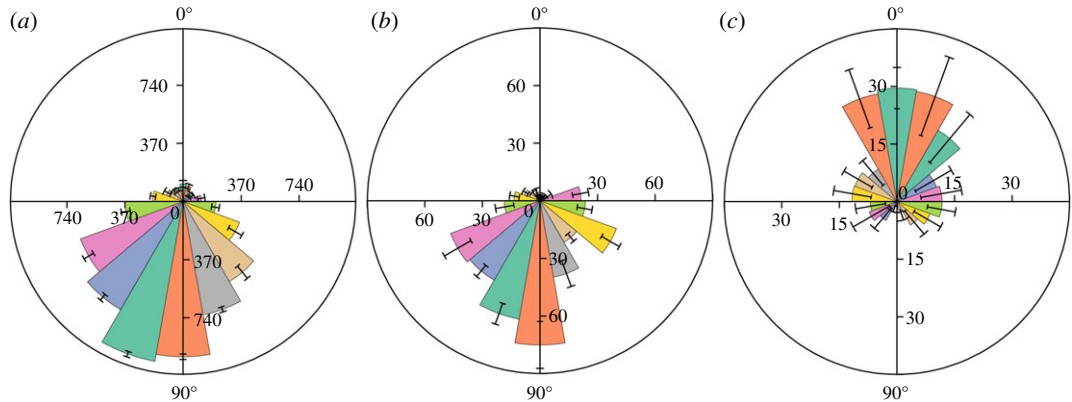

**Figure 10.** Distribution of crack plane orientation for tensile cracks in different simulation tests: (*a*) UCS, (*b*) Brazilian test and (*c*) direct tension test (10 numerical specimens with different bond strength distributions).

## 5.1. Loading condition

Pavlovskaia *et al*. [49] declared that a low-dimensional model (figure 11*c*) can well-simulate ultrasonic vibration systems. In this model, the ultrasonic dynamic loading is applied to the rock object through a block with mass $m_2$. The static loading force is directly applied to the block. This model was introduced into our study, and we need to use one kind of element to simulate both static and dynamic loadings.

In general, three common objects can be used to simulate loading conditions in PFC 5.0 (i.e. wall, ball and clump) [37,42]. The wall object does not obey the equations of motion. Thus, it can be only applied with a velocity, which means it is applicable for the static or quasi-static situation just as proposed in §4.2. For the dynamic situation, the ball and clump objects are usually used. They can not only be assigned with a velocity property but also a force condition, as these objects obey the equations of motion. However, when using the ball or clump, to decrease the geometry dispersion effect, increasing the wavelength/diameter is needed and this will greatly increase the number of balls and pebbles (the pebble is the basic element of the clump), further increasing the calculation time. Furthermore, each pebble of a clump is considered as a piece, often resulting in multiple contacts between a multi-pebble

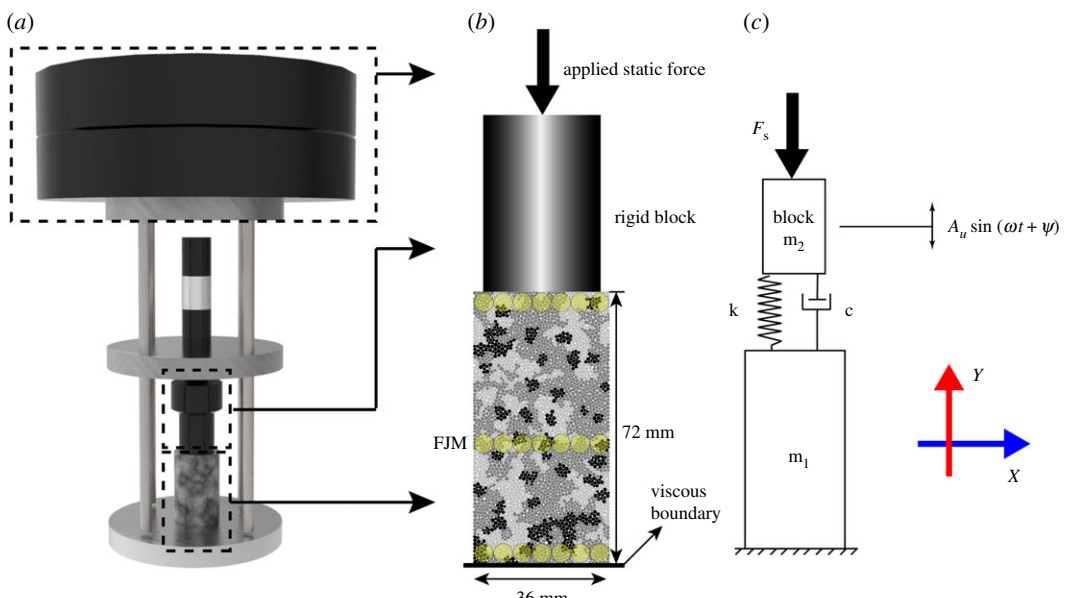

**Figure 11.** Loading method in the PFC simulations: (*a*) schematic diagram of experimental device; (*b*) schematic diagram of loading condition and layout of measurement circles (the yellow circles are measurement cycles and some of them are concealed) and (*c*) physical model of the UVBR system ($A_u$ is the velocity amplitude).

clump and another piece. Such an increase in the number of contacts can also be detrimental to computer performance.

In PFC 6.0, a new object called the rigid block (rblock) is presented [50]. Apart from the basic characteristics of the ball and clump, the rblock can directly model a convex object and does not need to use several particles. Thus, in the present work, one rblock element was used to simulate the loading plate, which would greatly decrease the calculation time.

The applied numerical loading strategy is the same as that used in the laboratory tests. A vertical force (200 N) is assigned to the rblock to simulate static loading. For simulating the dynamic loading, the rblock is assigned a velocity condition, and it is set as follows:

$$v_y = D_{Au} \cdot 2\pi f \cos(2\pi ft), \tag{5.1}$$

where $D_{Au}$ is the displacement amplitude of the ultrasonic vibration, m; $f$ is the loading frequency, Hz; and $t$ is the loading time, s.

## 5.2. Viscous boundary

In the modelling of the UVBR process, a viscous boundary should be applied along the bottom of the sample (figure 11*b*) to prevent failure due to the reflection of a wave from the boundary, especially in the high-frequency loading simulation [51]. As there is no built-in viscous boundary in PFC, a possible solution is to release the boundary balls in each timestep via the FISH function [52].

When a wave hits the boundary, a symmetric wave would be generated that could counteract the incoming one to simulate the viscous boundary. This symmetric wave is applied with minus equivalent stresses as the incoming one. The applied stress should be equivalent to the force in the DEM simulation. For the random distribution pattern, the equivalent expression between force and stress is

$$\begin{cases} \sum F_n = \beta_p \cdot D \cdot C \cdot \rho \cdot \dot{u}_{n-\text{body}} \\ \sum F_t = \beta_s \cdot D \cdot C \cdot \rho \cdot \dot{u}_{t-\text{body}} \end{cases}, \tag{5.2}$$

where $\sum F_n$ and $\sum F_t$ are the resultant normal and tangential contact forces of the element, N; $D$ is the diameter of the element, m; $\beta_p$ and $\beta_s$ are the normal and tangential inhomogeneous coefficients. To simplify the calculation, $\beta_p$ and $\beta_s$ are set to the same value, $\beta$; $\rho$ and $C$ are the medium's density, kg m$^{-3}$, and wave velocity, m s$^{-1}$.

The velocities of three balls in the top, medium and bottom parts of the sample were monitored to measure the wave velocity and obtain the absorption effects. From the measurement, $C$ is equal to

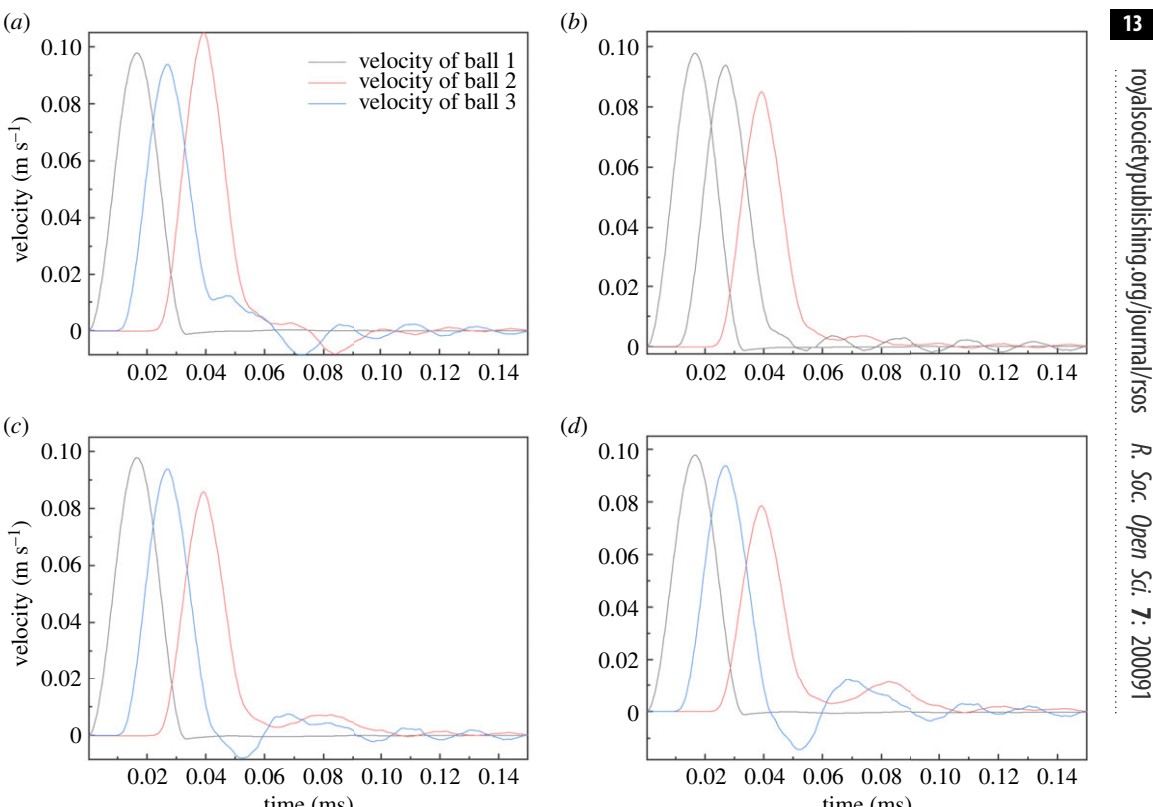

**Figure 12.** Velocity of three monitored balls with different inhomogeneous coefficients $\beta$: (a) 0.4, (b) 0.5, (c) 0.6 and (d) 0.7.

4300 m s$^{-1}$ in the model. The velocities of the three balls with $\beta$ of 0.4, 0.5, 0.6 and 0.7 are shown in figure 12. It can be found from the figure that when $\beta$ is 0.5, the absorption efficiency is optimum.

## 5.3. Attenuation characteristic of ultrasonic vibration

As shown in figure 6, local fragments generate in the UVBR process, and the evident failure occurs on the top of the sample, while the lower part is relatively complete. As an acoustic wave radiates from its source, geometric diffusing reduces the energy per unit area, which is related to the distance from the source. Since the amplitude of the acoustic wave is proportional to the square root of the energy, the pulse amplitude decreases with the distance from the source. Additionally, as the relative loss of energy per cycle is a constant, a pulse with a higher frequency would attenuate more rapidly than a pulse with a lower one, leading to a decent decrease in the pulse amplitude with increasing propagation distance [16]. Thus, it is essential to introduce the attenuation characteristic into the numerical model. One simple and effective way to realize this property is applying a damp property to the particles. The numerical trial-and-error method was used to determine the value of particle damp based on the strain amplitude measurement experiments.

In this experiment, four strain gauges were adhered to the surface of the rock sample in the axial direction with heights of 10, 27, 44 and 61 mm. A specific value of the strain amplitude obtained at different heights is $1 : 0.83 : 0.7 : 0.57$. This value is used to calibrate the damp parameter in the simulation. After calibration, the particle damp is chosen as 0.5. The numerical specific value of the strain amplitude in different heights is $1 : 0.85 : 0.73 : 0.61$.

## 5.4. Fatigue damage model

As mentioned in §4, the FJM can reproduce the heterogeneity and macro-properties of hard rock. However, it is unable to capture the fatigue damage growth in rock materials. This issue has indeed limited the applicability of PFC for modelling fatigue cracking. To overcome this limitation, a reduction model was proposed by Liu *et al*. [53] to simulate the fatigue behaviour of an intermittently jointed rock sample under low-frequency uniaxial compression. However, as there are many differences between the

(a) (b)

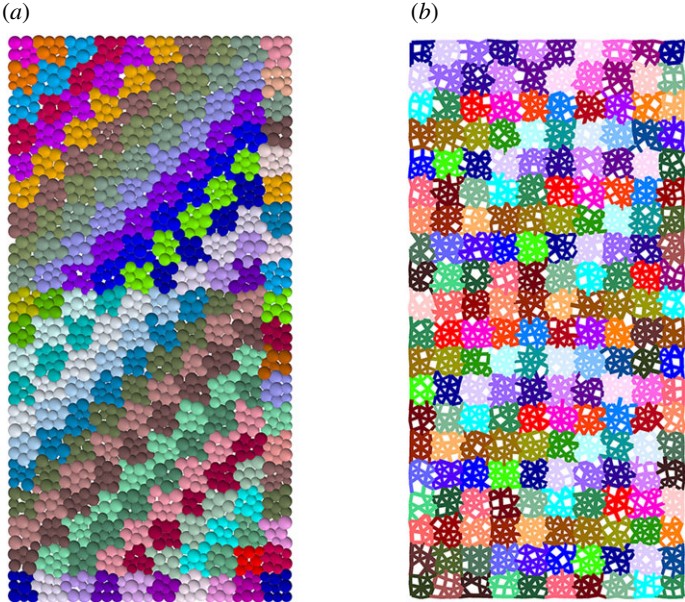

**Figure 13.** Different parts of the model: (a) particle groups and (b) contact groups.

low- and high-frequency loading conditions (i.e. the attenuation characteristic would cause significant fatigue anisotropy, and the small-stress cyclic loading would first make rock sample more 'compacted'), it is not appropriate in the present work. Thus, this model should be updated to make it suitable for ultra-dynamic conditions. In the updated model, the fatigue anisotropy and stiffness characteristics of rock under ultrasonic vibration are considered. The fatigue damage model methodology is described in detail below.

### 5.4.1. Model formulation

According to Sadd *et al.*'s research [14], when a rock sample is subjected to the cyclic loading condition, during one cycle, due to the dissipation of energy, the unloading stress–strain path lies below the loading curve, and damage accumulates. Thus, the basic idea behind simulating the rock fatigue is to vary the bond strength and stiffness at different stages (the loading, unloading and reloading stages) [54]. For the ultrasonic vibration condition, rock fatigue is characterized by anisotropy; accordingly, we established the rock fatigue model under ultrasonic vibration as follows:

$$
E = \begin{cases} E_0 \times \left(\dfrac{u_l^i}{u_0^i}\right)^a, \dot{u} > 0 \\[2mm] E_0 \times \left(\dfrac{u_u^i}{u_0}\right)^b, \dot{u} < 0 \\[2mm] E_0 \times \left(\dfrac{u_{rl}^i}{u_0^i}\right)^c, \dot{u} > 0 \end{cases}
\quad
S = \begin{cases} S_0 \times \left(\dfrac{u_l^i}{u_0^i}\right)^d, \dot{u} > 0 \\[2mm] S_0 \times \left(\dfrac{u_u^i}{u_0^i}\right)^e, \dot{u} < 0 , \\[2mm] S_0 \times \left(\dfrac{u_{rl}^i}{u_0^i}\right)^f, \dot{u} > 0 \end{cases}
\tag{5.3}
$$

where $\dot{u} > 0$ and $\dot{u} < 0$ denote the loading and unloading stages, respectively; $E$ and $S$ are the effective modulus and bonding strength (i.e. cohesive strength and tensile strength), respectively; and $E_0$ and $S_0$ are the initial effective modulus and bonding strength. Considering the fatigue anisotropy, the established model is divided into 200 parts (both the particles and contacts), as shown in figure 13. $u_0^i$ is the initial height of the $i$th part of the rock sample ($i$ is from 1 to 200), and $u_l^i$, $u_u^i$ and $u_{rl}^i$ are the height of the $i$th part at the loading, unloading and reloading stages, respectively. To simplify the calculation, the stiffness coefficients (i.e. $a$, $b$ and $c$) and strength coefficients (i.e. $d$, $e$ and $f$) are set to the same values, which are represented by $m$ and $n$, respectively. These coefficients will be calibrated according to the experimental results.

### 5.4.2. Cycle jump technique

As seen in figure 5, rock fails after millions of cycles under ultrasonic vibration; thus, the UVBR simulations would require extensive computational time if modelling cycle by cycle. To reduce the calculation time, the 'cyclic-jump' technique was used. This technique was first presented by

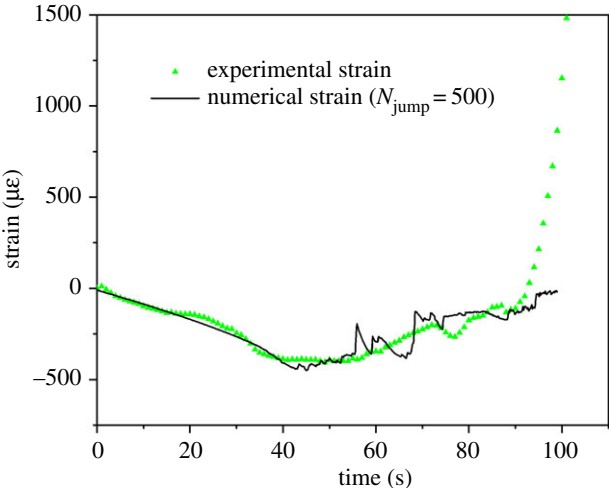

**Figure 14.** Experimental and simulated strain results (the simulation data are the average of the strain obtained from the measurement circles with id of 1, 8, 15 and 22 whose positions correspond to the strain gauge in the experiment).

Chaboche [55], and its assumption is that damage steadily develops in a small range of load cycles, which means that during that range, the simulation process can 'jump' from the first load cycle to the final load cycle. Additionally, the number of cycles omitted in a 'cycle jump' is determined according to the following equation:

$$N_{\text{jump}} = \frac{\delta D_{\text{jump}}}{\delta D} \tag{5.4}$$

where $N_{\text{jump}}$ is the number of omitted cycles, $\delta D_{\text{jump}}$ is an estimated damage increment after a cycle jump, and $\delta D$ is the average damage increment in the cycle jump range.

The damage increment in the present study is set as the ratio of the residual strain to the maximum residual strain. To simplify the calculation, $N_{\text{jump}}$ is set to 500, and in each second, the cycle number can be reduced to 60 cycles (we call them the 'simplified cycles' in the following), which saves considerable computational time. Figure 5 shows that the residual strain rate is approximately 10 $\mu\varepsilon\,\text{s}^{-1}$, which means that the strain would only change 0.16 $\mu\varepsilon$ per simplified cycle; thus, this simplification is reliable [56].

### 5.4.3. Fatigue model calibration

The fatigue damage model was calibrated by adjusting parameters (i.e. $m$ and $n$) until the numerical strain result matches the experimental one. After calibration, $m$ and $n$ are set as −0.4 and 1.1, respectively. Figure 14 depicts the dynamic strain curves obtained from the numerical simulations and the experiments. A reasonable consistency can be observed; although in the last stage, compared with the experimental result, the numerical strain value is relatively small. This difference is because the fragments generated in the simulation may not be separated from the matrix, as they actually are in the experiment.

The failure pattern in the numerical simulation is shown in figure 15. The irregular blocks with different colours represent fragments, and the azury ones are the matrix of the sample. The fragmentation process is similar to that in the experiments, with fragments first occurring at the top corner (in this case in the top-left corner) and then propagating to the adjacent area. In addition, with the increase in cycles, the fragment generation rate increases, which is also consistent with the experimental result.

## 6. Dynamic mechanical behaviour and the microcracking process

### 6.1. Distribution of stress and strain fields

One advantage of numerical simulations compared with laboratory tests is that they can directly detect the stress and strain fields. In this section, both the stress and strain fields were obtained based on the measured circle logics shown in figure 11*b*. There are 7 × 14 circles fully covering the rock sample area.

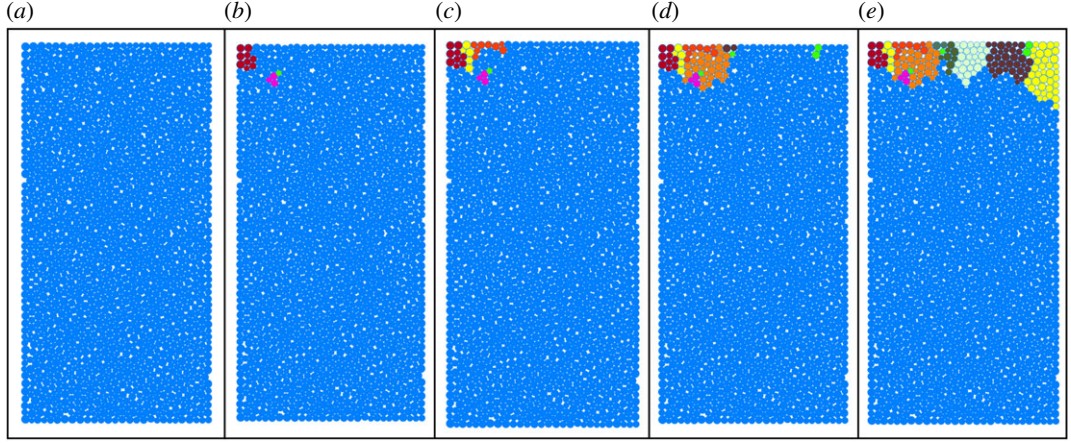

**Figure 15.** Development of the fragments: (*a*) 0, (*b*) 45, (*c*) 62, (*d*) 95 and (*e*) 127 s.

**Figure 16.** The distribution of stress in numerical sample under the ultrasonic vibration (*a*–*h* represent 4–120 s).

### 6.1.1. Stress field

Figure 16 illustrates the distribution of stress under ultrasonic vibration. It can be found that the stress increases with the calculation time, and the stress concentration also becomes evident. First, the

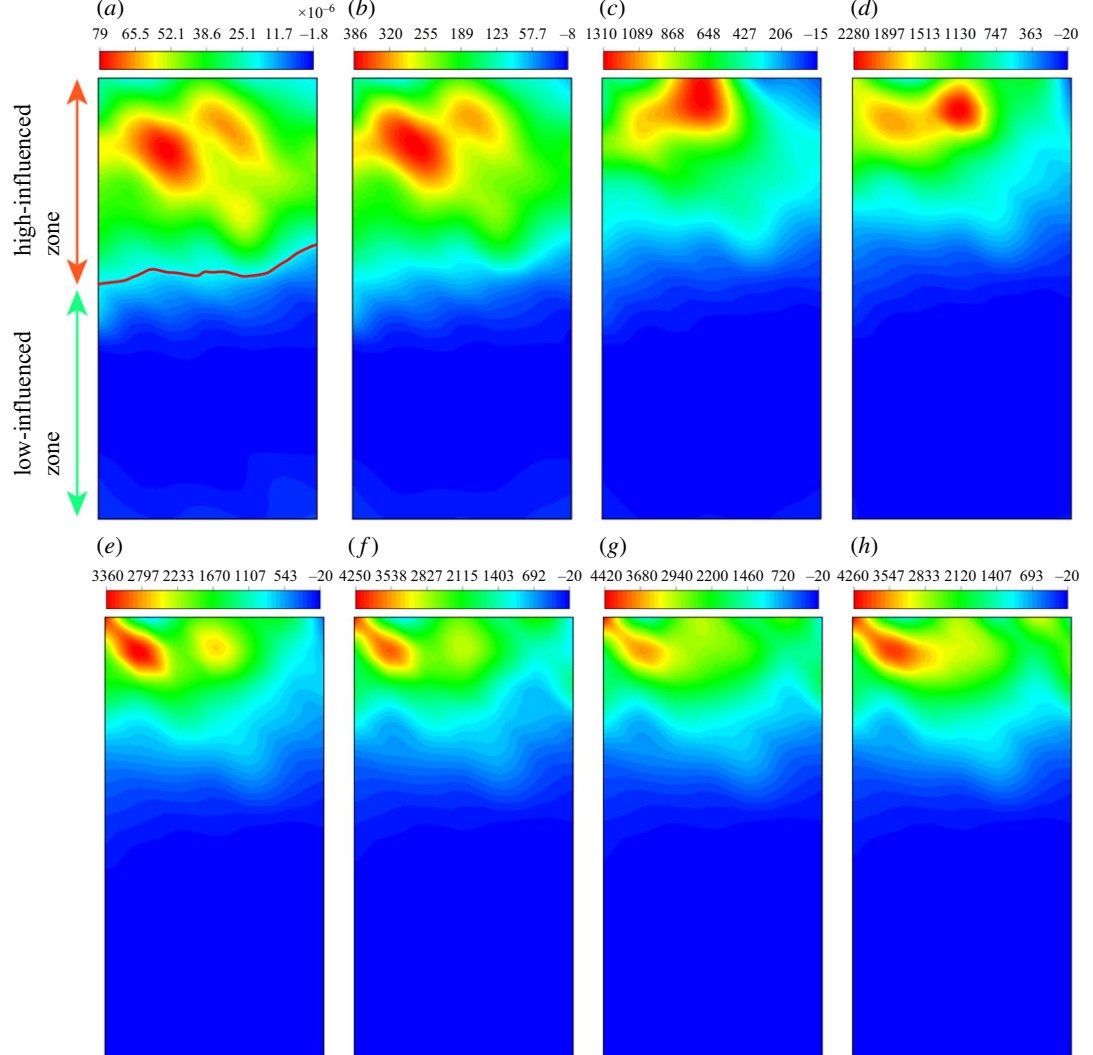

**Figure 17.** The distribution of x-strain in numerical sample (the negative value represents compressive strain, and the positive value represents tensile strain (a–h represent 4–120 s)).

stress concentration occurs at the top-left corner, then moves to the centre and at last to the right. As a result, the whole stress field is divided into three layers and the stress concentration is located at the junction of the first and second layers. This behaviour is because the initial fragments at the top-left corner cause a new significant stress concentration. As fragments continue to emerge, the stress concentration moves accordingly. The development of the stress field corresponds to the generation of fragments.

### 6.1.2. Strain field

As the failure process including the damage evolution, crack nucleation and propagation can be well described by the development of a strain field [57], the strain fields during the different loading stages were obtained.

Figure 17 shows the development of the x-direction strain field. As shown in figure 17a,b, after a period of loading, the regions of strain concentration are first localized in the middle and upper regions of the specimen. The strain concentration in this range is slight, which may be caused by the mechanical property differences between different minerals. Then, the concentration evolves into a narrower zone (figure 17c) and becomes significant. Such a localization of strain reflects the evolution of microcrack damage caused by compression during cyclic loading. Subsequently, the maximum value of local strain increases sharply from $13 \times 10^{-3}$ to $33 \times 10^{-3}$, and the strain at the top-left corner

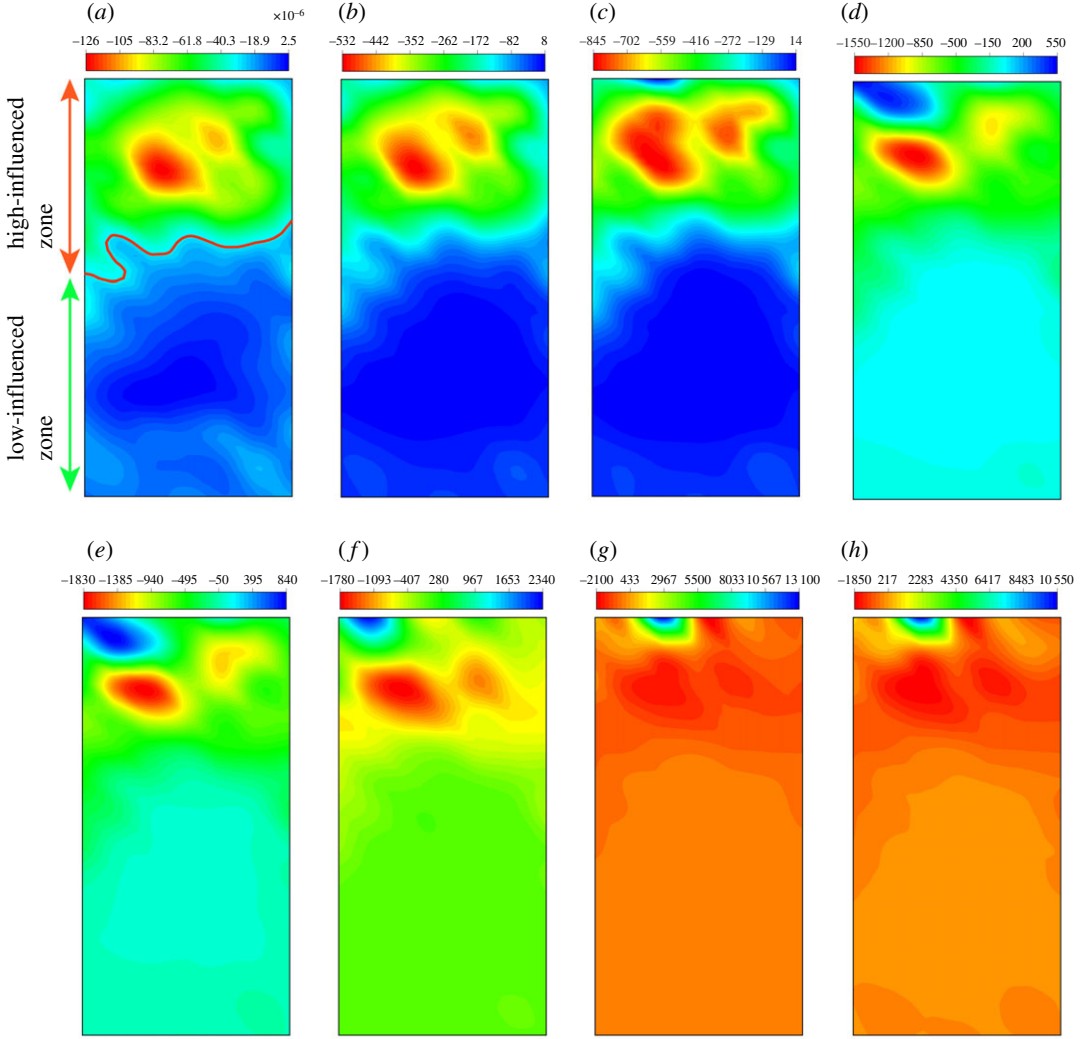

**Figure 18.** The distribution of y-strain in numerical sample (the negative value represents compressive strain, and the positive value represents tensile strain (a–h represent 4–120 s)).

also increases evidently, which indicates that the fragments are generated in this area (figure 17d,e). With the further increase in cyclic number, the strain field remains stable (figure 17f–h). The y-strain field develops similar to the x-direction one (figure 18).

## 6.2. Microcrack evolution

The acoustic emission (AE) counts and the development of damage and fragments used to declare the cracking process are shown in figure 19. The AE counts in the present work represent the generated crack number in each 50 timesteps, as Hunt *et al.* [58] found that the number of microcracks in the PFC2D is analogous to the AE count. The fragment in the PFC represents one block that contains one or more balls without bonded contacts in its boundary [42]. The mentioned damage is quantified by the following equation:

$$\eta = \frac{\sum_{i=1}^{n} r_i l_i}{\sum_{i=1}^{N} r_i l_i},$$  (6.1)

where $N$ and $n$ are the total crack number at ultimate failure and current crack number cumulated, respectively; $r_i$ and $l_i$ are the radius and length of the $i$th crack. The moment when the calculation is over corresponds to the damage state $\eta = 1$. The initial damage state is $\eta = 0$.

Based on the characterization of AE counts, damage and fragments, the fatigue process of rock under ultrasonic vibration can be divided into five stages (stages 1–5). In stage 1, no cracks occur. From

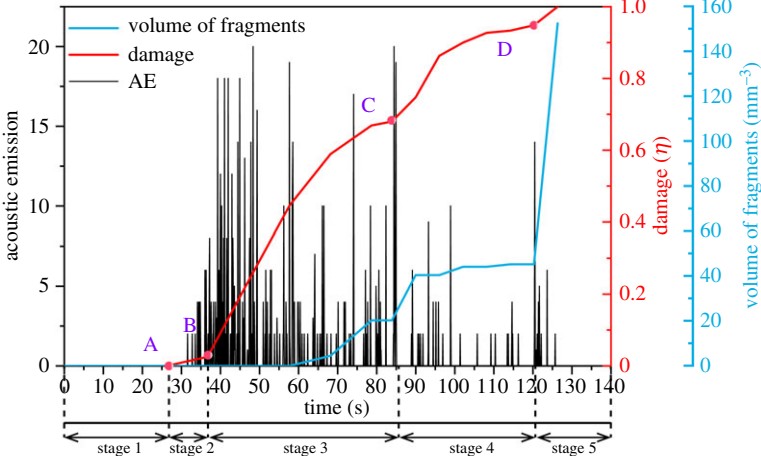

**Figure 19.** Various variables, AE counts (black bars), damage (red line) and volume of fragments (blue line) and changes as a function of vibration time.

point A, microcracks are generated corresponding to the crack initiation stage (stage 2). From point B, the cracks develop at a rate of 11.8 s⁻¹ and are followed by the generation of fragments, which corresponds to stage 3. In stage 4, the new cracks generate slowly, and a few fragments develop. Point D represents the final failure state, and the generation rate of fragments increases to 17.8 mm³ s⁻¹ during stage 5.

To further detail the development of cracks in each stage during the UVRB process, the damage in different position of rock sample (local damage) was also monitored. The local damage is calculated by equation (6.2), and its distribution is shown in figure 20d, which is obtained by dividing the sample area into $10 \times 20$ parts.

$$\eta_k = \frac{\sum_{i=1}^{n_k} r_i l_i}{\sum_{i=1}^{N_k} r_i l_i},$$
(6.2)

where $N_k$ and $n_k$ are the total crack number in the most damaged area at ultimate failure and the current crack number cumulated in the $k$th part of the sample, respectively. The area where the damage is severest at the moment when the calculation is over corresponds to the damage state $\eta_k = 1$. The initial damage state in each part is $\eta_k = 0$. Simultaneously, combining the microcracks' orientation distribution and type in figure 20a–c, we can summarize the cracking process as follows:

(1) No-crack stage. In this stage, the tensile $x$-strain gradually increases; however, it has not reached the crack initiation threshold, so no cracks are generated.
(2) Crack initiation stage. Initial microcracks are distributed at the top-left region, where the tensile $x$-strain is concentrated. All of the features are tensile cracks that result from compression. These crack orientations are limited to the range $70° < \theta < 110°$, which includes parallel and subparallel to the loading direction.
(3) Accelerating crack stage. It can be seen from the AE frequency that the microcracks are induced at a rate of 11.8 s⁻¹. The orientations of the tensile cracks are the same as those of the previous stage, that is parallel and subparallel to the loading direction. Additionally, the cracks gradually propagate from the top-left area to the right area. As some cracks coalesce with each other, the fragments begin to form.
(4) Stable stage. Influenced by the effective region, the cracks are generated slowly and at a rate of 7 s⁻¹. The upper part of the sample is almost covered by the cracks. The effective region is approximately completely destroyed, which means that there is little area to generate new microcracks. Apart from the tensile cracks, a few shear cracks are also generated. Additionally, the tensile cracks' orientation distribution has become more homogeneous.
(5) Final failure stage. With the further increase in strain, new cracks begin to connect with other cracks to generate a shear band, which results in a sharp increase in fragments, and some secondary cracks are generated accordingly, which increase the damage. The significant damage area increases. The shear microcrack orientations are limited to the ranges $0° < \theta < 30°$ and $150° < \theta < 180°$, which correspond to vertical and subvertical to the loading direction.

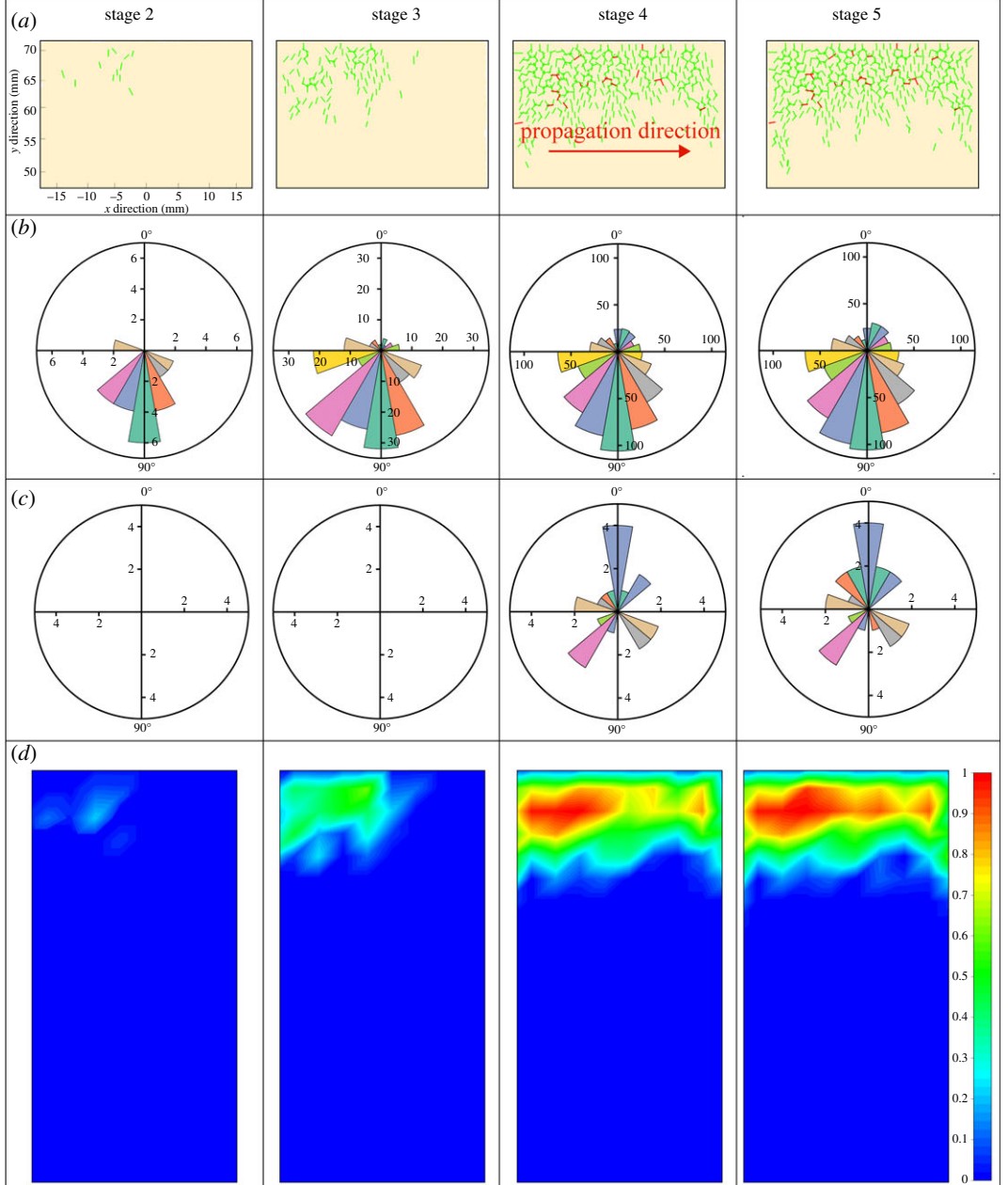

**Figure 20.** Fracture initiation and propagation and damage evolution process during UVBR process: (*a*) DEM simulated cracks' distribution and propagation, (*b*,*c*) corresponding crack orientation, and (*d*) corresponding damage field. Different columns represent different stages 2–5. Green lines represent tensile cracks and red shear cracks.

Based on the analysis of the numerical and experimental results described above, the mechanism for the formation of cracks under ultrasonic vibration is revealed. For the tensile cracks (figure 21*a*), as the compressive *y*-strain and corresponding tensile *x*-strain increase, two connected particles part from each other in the *x*-direction, which causes the contact between the two particles (particles 2 and 3) to experience tensile strain. As the loading continues, the tensile strain increases, and the tensile strain limitation, which corresponds to the contact bond strength between particles [59], decreases because of the generation of fatigue damage of rock materials under ultrasonic vibration loading. When the tensile strain exceeds the tensile strain limitation, the crack perpendicular to the *x*-axis forms. For the shear cracks (figure 21*b*), as the strain concentration becomes more evident, the strain difference between the strain concentration area and its surrounding area becomes larger. Due to the strain difference, particles (particle 1 and particles 2 and 3) at the interface produce a strong dislocation, which finally leads to the generation of shear cracks.

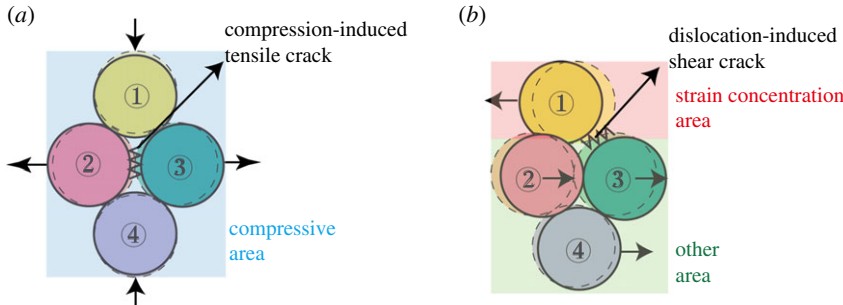

**Figure 21.** Crack mechanism of rock under ultrasonic vibration: (*a*) compression-induced tensile crack and (*b*) dislocation-induced shear crack.

# 7. Conclusion

In this study, we investigated the fatigue behaviour and crack evolution mechanics of granite under ultrasonic vibration with experimental tests and discrete element simulations. Based on the obtained experimental and numerical results, the following conclusions can be drawn.

(1) Tested intact granite is a typical brittle rock that undergoes obvious local failure (upper part of the sample) under ultrasonic vibration. The experimental strain curves clearly depict three stages. During the first stage, the rock experiences compressive deformation caused by its nonlinear property. Crack closure occurs and there is no apparent rupture. As the vibration continues, the compressive deformation decreases due to the initiation and propagation of cracks. Finally, with the further of crack initiation, propagation and coalescence, large fracture forms at the edge and then propagates to the adjacent area.

(2) During the early stage, the stress amplitude of the sample gradually increases due to the increase in the natural frequency and modulus caused by the compressive deformation and corresponding crack closure. Thus, the rock fatigue will accelerate due to the increase in the stress amplitude, and this will further accelerate the rock-breaking process.

(3) To understand the crack evolution mechanism of brittle granite specimens under ultrasonic vibration, numerical simulations by PFC2D were performed, and an optimal fatigue FJM was proposed, which could not only highly reproduce the properties of hard brittle rock but also capture the fatigue damage characteristic under high-frequency loading. The numerical simulation results show good agreement with the experimental results in terms of the development of fragments and strain. By analysing the microcrack orientation distribution and crack type, crack evolution was found to include five stages, namely, the undamaged stage, crack initiation stage, accelerated cracking stage, stable stage and final failure stage.

(4) Based on the strain fields obtained from simulations, the mechanism for the formation of cracks under ultrasonic vibration is revealed: the compression-induced tensile cracks are generated due to the separation of particles resulting from the increase in axial compressive strain, while the dislocation-induced shear cracks were generated due to the increase in axial strain difference between the strain concentration area and its surrounding area.

Data accessibility. The FISH code used for establishing clustered rock model for this research work is stored in GitHub: https://github.com/Python-Zhou/PFC-public.git and have been archived within the Zenodo repository: http://doi.org/10.5281/zenodo.3626246.

Authors' contributions. Y.Z. carried out the laboratory work, participated in data analysis, participated in the programme and drafted the manuscript; Q.T. conceived of the study, participated in the design of the study and drafted the manuscript; D.Z. revised work critically and coordinated the study; M.W. carried out data analysis and interpretation. All authors gave final approval for publication.

Competing interests. The authors declare that they have no conflict of interest.

Funding. This work was supported in part by the National Natural Science Foundation of China (grant no. 4157020248) and in part by Graduate Innovation Fund of Jilin University (grant no. 101832018C046), which is greatly appreciated.

Acknowledgements. Our deepest gratitude goes to the anonymous reviewers for their careful work and thoughtful suggestions that have helped improve this paper substantially.

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
