## [Reviewer comments · Royal Society Open Science]

Review History

RSOS-200091.R0 (Original submission)

Review form: Reviewer 1

Is the manuscript scientifically sound in its present form?

Yes

Are the interpretations and conclusions justified by the results?

Yes

Is the language acceptable?

Yes

Do you have any ethical concerns with this paper?

No

Have you any concerns about statistical analyses in this paper?

No

Recommendation?

Major revision is needed (please make suggestions in comments)

Comments to the Author(s)

- 1) The introduction part needs to be well organized again, and it should focus on the content related to this study.
- 2) One dedicated paragraph must be there for explaining the research gap. The motivation for the work should spontaneously come out from the research gap itself which is not evident.
- 3) It is necessary to give the complete stress-strain curve of the sample and explain why the static load adopted is 200N (Whether 200N reaches the fatigue triggering condition).
- 4) "Thus, to improve the fragmentation efficiency of this technology, the applied ultrasonic vibration frequency should be changed in real time to approach the natural frequency of the rock." Is resonance the essence of this sentence?
- 5) "A specific value of the strain amplitude obtained at different heights is 1:0.83:0.7:0.57." No corresponding strain gauge is found in the experimental picture. What is the relationship between strain and damping? How to deal with it?
- 6) PFC numerical simulation modeling process should be simplified, part of which is the common sense of software users.
- 7) How to verify the correctness of numerical simulation results? Please give the stress-strain curve of the simulation result and compare it with the stress-strain curve of the sample.
- 8) Please rearrange the conclusions of this paper and emphasize the innovative work.

Review form: Reviewer 2 (John Browning)

Is the manuscript scientifically sound in its present form?

No

Are the interpretations and conclusions justified by the results?

Yes

Is the language acceptable?

Yes

Do you have any ethical concerns with this paper?

No

Have you any concerns about statistical analyses in this paper?

No

Recommendation?

Major revision is needed (please make suggestions in comments)

Comments to the Author(s)

This manuscript presents new data from a suite of uniaxial compression and Brazilian disc experiments that utilize an ultrasonic vibration device to apply ultra-high frequency loads to the samples of fine grained granite. Some of the experimental data was used to calibrate a suite of PFC numerical simulations which were in turn used to explain observations from the ultra-high-frequency loading tests. The numerical simulations incorporate a new flat-joint method of coupling bonds which reduces or eliminates issues related to the more commonly used parallel-bond method. The experimental results demonstrate three main phases of deformation resulting from the ultrasonic vibrations, discerned from a single strain gauge positioned at the upper most part of the sample. The numerical results are more numerous and this seems the main focus of the paper in reality. Resultant stress and strain fields are reproduced in the models and Acoustic

Emission output is also considered as a proxy for crack damage. It seems strange to me that the authors did not monitor Acoustic Emission output in the experiments as this would provide an even more robust check of the numerical data and especially permit confirmation of fatigue or damage memory effects. In all I found the descriptions of the experimental methods and results very difficult to follow and I suggest these parts require substantial re-writing. I also thought that the Introduction did an inadequate job of explaining the motivation for the study and the section felt quite disjointed the way it has been written. Again, I suggest this section be reworked.

As regards the scientific content, I found the study to be interesting although I have some concerns regarding the experimental setup. To start, what was the motivation for using just one single strain gauge? It is not explicitly clear in the manuscript which component of strain is plotted and discussed throughout. I am concerned that by placing the strain gauge near the upper sample surface, and not recording the strain in the central part of the sample, some of the variety of processes occurring are being obscured. This may be problematic when also considering that the upper and lower surfaces of materials loaded in uniaxial conditions normally experience edge effects. This process is explained in detail in section 3.5 of the book 'Rock deformation the Brittle Field' by Paterson. It seems from the experiments and models that most of the fracture damage is confined to the upper most portion of the rock sample. Was any material placed between the piston and the sample to eliminate such effects?

Please try to avoid the use of qualitative terms such as 'small' and 'large' for describing the results or cycle numbers, for example. Such terminology can be avoided because the actual values can be given. I have made many comments on a marked up PDF version of the manuscript which I attach (Appendix A). Some of the more urgent comments that I briefly note here relate to Figure 5. Can the authors comment on why the amounts of strain and duration (strain rate) of the experiments are so variable? There is little comment in the manuscript about repeatability and checking this plot leads me to question how repeatable the tests are. On this point, I think a proper characterisation of the material used would be very welcome. There is some information about the grain size but not an examination of the distribution of pre-existing fractures. How heterogeneous or anisotropic is the granite? Where was it collected from etc?

Also, I'm not convinced that crack length is a suitable parameter to use in equation 1. Is it not crack opening or crack surface area that will control the frequency more than simply length?

Overall I feel that the manuscript requires major revision and substantial rewriting before it is reconsidered for publication in Royal Society Open Science.

Decision letter (RSOS-200091.R0)

26-Feb-2020

Dear Dr Zhou,

The editors assigned to your paper ("Experimental and Numerical Investigation of the Fatigue Behaviour and Crack Evolution Mechanism of Granite Under Ultra-High-Frequency Loading") have now received comments from reviewers. We would like you to revise your paper in accordance with the referee and Associate Editor suggestions which can be found below (not including confidential reports to the Editor). Please note this decision does not guarantee eventual acceptance.

Please submit a copy of your revised paper before 20-Mar-2020. Please note that the revision deadline will expire at 00.00am on this date. If we do not hear from you within this time then it will be assumed that the paper has been withdrawn. In exceptional circumstances, extensions

may be possible if agreed with the Editorial Office in advance. We do not allow multiple rounds of revision so we urge you to make every effort to fully address all of the comments at this stage. If deemed necessary by the Editors, your manuscript will be sent back to one or more of the original reviewers for assessment. If the original reviewers are not available, we may invite new reviewers.

- Data accessibility

If you wish to submit your supporting data or code to Dryad (<http://datadryad.org/>), or modify your current submission to dryad, please use the following link:
<http://datadryad.org/submit?journalID=RSOS&manu=RSOS-200091>

- Competing interests

- Authors' contributions

- Acknowledgements

- Funding statement

on behalf of Dr Philip Benson (Associate Editor) and R. Kerry Rowe (Subject Editor)
openscience@royalsociety.org

Reviewers' Comments to Author:

Reviewer: 1

Comments to the Author(s)

1) The introduction part needs to be well organized again, and it should focus on the content related to this study.

2) One dedicated paragraph must be there for explaining the research gap. The motivation for the work should spontaneously come out from the research gap itself which is not evident.

3) It is necessary to give the complete stress-strain curve of the sample and explain why the static load adopted is 200N (Whether 200N reaches the fatigue triggering condition).

4) "Thus, to improve the fragmentation efficiency of this technology, the applied ultrasonic vibration frequency should be changed in real time to approach the natural frequency of the rock." Is resonance the essence of this sentence?

5) "A specific value of the strain amplitude obtained at different heights is 1:0.83:0.7:0.57." No corresponding strain gauge is found in the experimental picture. What is the relationship between strain and damping? How to deal with it?

6) PFC numerical simulation modeling process should be simplified, part of which is the common sense of software users.

7) How to verify the correctness of numerical simulation results? Please give the stress-strain curve of the simulation result and compare it with the stress-strain curve of the sample.

8) Please rearrange the conclusions of this paper and emphasize the innovative work.

Reviewer: 2

Comments to the Author(s)

This manuscript presents new data from a suite of uniaxial compression and Brazilian disc experiments that utilize an ultrasonic vibration device to apply ultra-high frequency loads to the samples of fine grained granite. Some of the experimental data was used to calibrate a suite of PFC numerical simulations which were in turn used to explain observations from the ultra-high-frequency loading tests. The numerical simulations incorporate a new flat-joint method of coupling bonds which reduces or eliminates issues related to the more commonly used parallel-bond method. The experimental results demonstrate three main phases of deformation resulting from the ultrasonic vibrations, discerned from a single strain gauge positioned at the upper most part of the sample. The numerical results are more numerous and this seems the main focus of the paper in reality. Resultant stress and strain fields are reproduced in the models and Acoustic Emission output is also considered as a proxy for crack damage. It seems strange to me that the authors did not monitor Acoustic Emission output in the experiments as this would provide an even more robust check of the numerical data and especially permit confirmation of fatigue or damage memory effects. In all I found the descriptions of the experimental methods and results very difficult to follow and I suggest these parts require substantial re-writing. I also thought that the Introduction did an inadequate job of explaining the motivation for the study and the section felt quite disjointed the way it has been written. Again, I suggest this section be reworked.

As regards the scientific content, I found the study to be interesting although I have some concerns regarding the experimental setup. To start, what was the motivation for using just one single strain gauge? It is not explicitly clear in the manuscript which component of strain is plotted and discussed throughout. I am concerned that by placing the strain gauge near the upper sample surface, and not recording the strain in the central part of the sample, some of the variety of processes occurring are being obscured. This may be problematic when also considering that the upper and lower surfaces of materials loaded in uniaxial conditions normally experience edge effects. This process is explained in detail in section 3.5 of the book 'Rock deformation the Brittle Field' by Paterson. It seems from the experiments and models that most of the fracture damage is confined to the upper most portion of the rock sample. Was any material placed between the piston and the sample to eliminate such effects?

Please try to avoid the use of qualitative terms such as 'small' and 'large' for describing the results or cycle numbers, for example. Such terminology can be avoided because the actual values can be given. I have made many comments on a marked up PDF version of the manuscript which I attach. Some of the more urgent comments that I briefly note here relate to Figure 5. Can the authors comment on why the amounts of strain and duration (strain rate) of the experiments are so variable? There is little comment in the manuscript about repeatability and checking this plot leads me to question how repeatable the tests are. On this point, I think a proper characterisation of the material used would be very welcome. There is some information about the grain size but not an examination of the distribution of pre-existing fractures. How heterogeneous or anisotropic is the granite? Where was it collected from etc?

Also, I'm not convinced that crack length is a suitable parameter to use in equation 1. Is it not crack opening or crack surface area that will control the frequency more than simply length?

Overall I feel that the manuscript requires major revision and substantial rewriting before it is reconsidered for publication in Royal Society Open Science.

Author's Response to Decision Letter for (RSOS-200091.R0)

See Appendix B.

Decision letter (RSOS-200091.R1)

23-Mar-2020

Dear Dr Zhou,

It is a pleasure to accept your manuscript entitled "Experimental and Numerical Investigation of the Fatigue Behaviour and Crack Evolution Mechanism of Granite Under Ultra-High-Frequency Loading" in its current form for publication in Royal Society Open Science.

on behalf of Dr Philip Benson (Associate Editor) and R. Kerry Rowe (Subject Editor)
openscience@royalsociety.org

Appendix A**ROYAL SOCIETY
OPEN SCIENCE****Experimental and Numerical Investigation of the Fatigue
Behaviour and Crack Evolution Mechanism of Granite Under
Ultra-High-Frequency Loading**

Journal:	Royal Society Open Science
Manuscript ID	RSOS-200091
Article Type:	Research
Date Submitted by the Author:	24-Jan-2020
Complete List of Authors:	Zhou, Yu; Jilin University, Zhao, Dajun; Jilin University Tang, Qiongqiong; Jilin University Wang, Meiyang; Jilin University; Shandong Vocational College of Science and Technology
Subject:	Computer modelling and simulation < COMPUTER SCIENCE, Civil engineering < ENGINEERING AND TECHNOLOGY, Geology < EARTH SCIENCES
Keywords:	Ultrasonic vibration, Granite, Fatigue behaviour, PFC2D, Crack evolution, Fatigue damage model
Subject Category:	Engineering

Author-supplied statements

Relevant information will appear here if provided.

Ethics

Does your article include research that required ethical approval or permits?:

This article does not present research with ethical considerations

Statement (if applicable):

CUST_IF_YES_ETHICS :No data available.

Data

It is a condition of publication that data, code and materials supporting your paper are made publicly available. Does your paper present new data?:

Yes

Statement (if applicable):

<https://github.com/Python-Zhou/PFC-public.git>

<http://doi.org/10.5281/zenodo.3626246>.

Conflict of interest

I/We declare a competing interest

Statement (if applicable):

CUST_STATE_CONFLICT :No data available.

Authors' contributions

This paper has multiple authors and our individual contributions were as below

Statement (if applicable):

Funding acquisition, Yu Zhou, Qiongqiong Tang and Dajun Zhao; Investigation, Qiongqiong Tang and Yu Zhou; Methodology, Qiongqiong Tang; Software, Qiongqiong Tang, Meiyan Wang and Yu Zhou; Supervision, Dajun Zhao and Yu Zhou; Visualization, Yu Zhou; Writing " original draft, Qiongqiong Tang; Writing " review & editing, Dajun Zhao and Yu Zhou.

Experimental and Numerical Investigation of the Fatigue Behaviour and Crack Evolution Mechanism of Granite Under Ultra-High-Frequency Loading

YU ZHOU¹, DAJUN ZHAO¹, QIONGQIONG TANG¹, MEIYAN WANG^{1,2}

¹ Complex Condition Drilling Experiment Center, Jilin University, 130012 Changchun, China

² Shandong Vocational College of Science and Technology, 261000 Weifang, China

Corresponding author: Meiyang Wang (9101037@qq.com) and Qionqiong Tang (tangqq17@mails.jlu.edu.cn)

This work was supported in part by the National Natural Science Foundation of China under Grant 4157020248, and in part by Graduate Innovation Fund of Jilin University under Grant 101832018C046

ABSTRACT Assisted ultrasonic vibration technology has received great interest in the past few years for petroleum and mining engineering related to hard rock breaking. Understanding the fatigue behaviour and damage characteristics of rock subject to ultra-high-frequency loading is vital for its application. In this research, we conducted ultrasonic vibration breaking rock experiments combined with an ultra-dynamic data receiver and strain gauges to monitor the development of strain in real time. The experimental results show that the strain curve first decreased, then remained steady and finally increased. The sample first underwent compressive deformation, and no rupture occurred. As the vibration continued, the compressive deformation decreased with the initiation and propagation of cracks, and significant fragmentation occurred. To elucidate the crack evolution mechanism of the granite specimens, numerical simulations were performed using PFC2D, and an improved fatigue damage model based on the flat-joint contact model was proposed. The numerical results indicate that this model can effectively reproduce the fatigue characteristics of hard brittle rocks under ultrasonic vibration. By analysing the stress and strain fields and cracking process, the crack evolution mechanism in brittle hard rock under ultra-high-frequency loading is revealed. These experimental and numerical results are expected to improve the understanding of the fragmentation mechanism of rock under assisted ultrasonic vibration.

INDEX TERMS Ultrasonic vibration · Granite · Fatigue behaviour · PFC2D · Crack evolution · Fatigue damage model

1. INTRODUCTION

Due to the high strength and wide distribution of hard rock, rapidly breaking hard rock for engineering projects, such as tunnelling and petroleum and geological drilling, is a widespread issue [1-3]. To quickly break hard rock, technological advancements have used the assisted ultrasonic vibration technique [4]. Uniaxial compressive strength experiments were conducted to obtain the development of rock strength after ultrasonic vibration by Yin et al.; the results demonstrated that this technique can effectively reduce the strength of hard granite and the effect of the static pressure on the efficiency of rock breaking was examined. There is a static pressure threshold for this technology above which fatigue damage would occur [5]. Zhao et al. divided the rock into three zones under ultrasonic vibration: the fracture zone, plastic deformation zone, and elastic deformation zone [6]. Zhou et al. investigated the effect of static loading on the natural

frequency of the system when subjected to ultrasonic vibration [7]. However, these researchers have not expanded their investigations to explore the fatigue behaviour and fragmentation mechanics of granite under ultrasonic vibration, which is of significance for utilizing this technique.

Fortunately, many investigations have focused on the fatigue and mechanical behaviour of rock under cyclic loading. A fatigue threshold stress was confirmed under triaxial cyclic loading [8]. The fatigue threshold stress is the stress level corresponding to the transition from volumetric compaction to volumetric dilation. In the low stress region, the stiffness of the rock will first increase and then decrease when crack growth occurs after repeated loading [9]. The energy evolution was investigated by Liu et al., who found that the total absorbed energy mainly converts into elastic energy. Thereafter, all the stored energy is released in the form of dissipated hysteresis energy for the propagation and

1
2
3
4
5
6
7
8
9
10
11
12
13
14
15
16
17
18
19
20
21
22
23
24
25
26
27
28
29
30
31
32
33
34
35
36
37
38
39
40
41
42
43
44
45
46
47
48
49
50
51
52
53
54
55
56
57
58
59
60

coalescence of microcracks [10]. Yang et al. [11] used the ultrasonic wave velocity method to assess sandstone damage and found that rock specimens stiffen during loading and soften during unloading. Jiang et al. [12] found that the porosity and pore size of marble rock gradually decreases under cyclic dynamic impact and that its framework becomes compacted. Apart from marble rock, red sandstone was also investigated by this research group, and the authors concluded that the damage variables showed a strong non-linear relationship with the number of dynamic impacts and that the damage increases with the number of dynamic impacts [13].

Clearly, the evolution of microcracks plays an essential role in the rock failure process. It is therefore necessary to investigate the growth of cracks during the fatigue process. However, the monitoring of crack growth development in dynamic experiments is difficult. With the rapid development of computer technology and advanced numerical techniques, more details, especially the characterization of crack growth, can be determined. For example, Hu et al. [14] used LS-DYNA to study the rock fragmentation mechanisms when cut by a cutter. Yang et al. [15] investigated the rock breaking mechanism under supercritical (SC)-CO₂ jet impacting based on the smoothed particle hydrodynamic finite element method (SPH-FEM). Yang et al. [16] investigated the influence of cracks on the mechanical properties of rock using Franc3D and found that the strength decreases with a decrease in the crack dip angle and an increase in the crack density. Rock failure process analysis (RFPA) was used to study the internal damage evolution of rock under uniaxial compression and tension [17]. Recently, the discrete element method (DEM) has received considerable attention for its advantages in simulating the microcrack evolution and discontinuity property of rock materials. He et al. proposed a new simplified model generation approach based on the Monte Carlo method to simulate the complicated structure of Rock-soil aggregate. Song et al. determined the failure position of a tunnel based on a DEM simulation and concluded that it is an efficient method to predict the instability in the rock surrounding a tunnel with a complex geological condition. Based on a novel clustered assembly approach, Li et al. [18] studied the rock fragmentation mechanism and failure process induced by a wedge indenter and found that the initial void and defects existing in the model exert a significant effect on the coalescence and propagation of a median crack as well as on the stress concentration. Tian et al. [19] verified that shear cracks are more difficult to initiate in coarse granite specimens than in fine granite specimens. A series of particle flow code in three dimensions (PFC3D) simulations on numerical specimens were carried out, and the results showed that different heights of the sample result in different failure patterns [20].

The abovementioned studies of rock fatigue behaviour and crack evolution were all under low-frequency cyclic loading or a quasi-static condition. However, ultrasonic vibration is characterized by its ultra-high frequency, which is on the same order of magnitude as the natural frequency of hard rock [21], low amplitude (frequency: ≥ 20 KHz; amplitude: 10-30 μm) and fast energy loss [22], and these characteristics are quite different from the traditional loading conditions. Thus, it is significant and essential to further study the fragmentation mechanisms of rock under ultrasonic vibration.

One aim of this study is to investigate the fatigue behaviour of granite under ultrasonic vibration based on the strain test using an ultra-dynamic data receiver and strain gauges in real time. Another aim is to utilize a two-dimensional discrete numerical code, PFC2D, to simulate the mesoscopic laboratory test and explore the crack evolution law in the process of ultrasonic vibration breaking rock (UVBR), which is difficult to observe in dynamic experiments. The long-term aim is to use the established model to examine the loading parameters (i.e., frequency and amplitude of a dynamic load and the magnitude of static loading) on the fragmentation efficiency and to know more about rock breaking mechanisms.

In this study, both physical and numerical ultrasonic vibration tests were conducted to systematically investigate the fatigue behaviour and crack evolution law of brittle rocks under ultra-high-frequency loading conditions. This paper is organized as follows. Section 2 introduces the experimental methodology, including the loading method and facility and the specimen used. Section 3 reports the experimental results of the ultrasonic vibration tests and analyses the fatigue behaviour of rock. In Section 4, details of the rock model generation and micro-parameter calibration procedure are presented. Section 5 proposes the numerical methodology to model the UVBR process, and it concludes with four key points. The mechanical fatigue behaviour and crack evolution results of the rock specimen is analysed and revealed in the following section. Finally, Section 7 summarizes the conclusions drawn from this study.

II. EXPERIMENTAL METHODS

A SAMPLE PREPARATION

The rock samples used in the experiments were processed from fine granite, a common type of hard brittle rock. Its mineralogical composition was determined by X-ray diffraction analysis (Fig. 1), which shows that this rock contains quartz, albite, orthoclase and hydrobiotite. The samples were shaped into standard cylindrical blocks with a diameter of 36 mm and a height of 72 mm. The average grain size is approximate 1.1 mm, as shown in Fig. 2. The approximate proportions of the different minerals are feldspar (29.4%), quartz (50.6%), orthoclase and

hydrobiotite (18.3%) and others (1.7%). According to the International Society of Rock Mechanics [23] standards, the sample size was assumed to be large enough to ignore its size effect. Simultaneously, to reduce the effect of the rock heterogeneity on the experimental results, samples were analysed using the knocking method to examine their natural frequencies. Samples with similar natural frequencies from 26 to 27 KHz were picked out, and one of their natural frequency test results is shown in Fig. 3.

B TESTING SETUP

The test equipment comprises an ultrasonic vibration device and a data acquisition device, as shown in Fig. 4a. The loading apparatus is composed of an ultrasonic power source (1), an ultrasonic vibrator (2) and a static loading device (3). The loading method in this test is shown in Fig. 4b. An ultrasonic dynamic load combined with a vertical static load were applied to the samples. The static load was applied with weights, and each of them weighs 10 N. They were placed on the top of the bracket. The data collection apparatus includes a computer (5) and an ultrasonic data receiver (4) combined with a strain gauge (the effective size is 20*3 mm) (6), and it was used to collect strain data in real time. The strain gauge was placed on the top side surface of the cylindrical rock specimen, where the damage was severest.

The static loading and ultrasonic vibration frequency are 200 N and 30 KHz, respectively, under which the crushing efficiency is higher based on the previous experiments [24]. The strain behaviour of the rock was monitored until obvious fragments occurred. All the experiments were conducted at room temperature.

III. EXPERIMENTAL RESULTS AND DISCUSSION

Figure 5 shows the development of axial strain for 6 samples (the red and green lines are the slope of the strain curve). Although the strain curves are not identical due to the heterogeneity of rock, it is easy to conclude the development law. The strain curve is U-shaped, and it can be divided into 3 stages: the strain first decreases, then remains steady (with small fluctuations), and finally increases. The negative strain in the figure represents compressive deformation, and the positive one represents tensile deformation. In the first stage, irreversible compressive deformation gradually increases, which would lead to crack closure and make the sample more 'compacted' [13, 25]. In other words, the modulus of rock would increase. As the vibration continues, the slope of the strain curve decreases. New cracks would gradually form, and the initial cracks would propagate and reopen. These crack behaviours would result in a decrease in rock modulus and cause rock expansion in the axial direction. As the cracks continue to propagate, the magnitude of the expansion would become larger. Thus, the strain increases in the last stage.

To observe the complete fragmentation process of the rock sample under ultrasonic vibration, we applied ultrasonic vibration to another undamaged sample, as shown in Fig. 6. First, a relatively small fragment zone occurs at the edge of the sample. Then, the fragment zone continues to extend from the initial region to the adjacent area, and the fragment zone becomes larger. Finally, macroscopic damage distributes over the whole top edge and further develops into the centre. According to the fragment pattern, it can be concluded that this technique covers an influential range.

Figure 7 illustrates the development in the strain amplitude. The strain amplitude is the difference between the maximum and the minimum strains in each second. The strain amplitude was obtained and combined with the development in the elastic modulus to track the stress amplitude, which can greatly affect the fatigue behaviour of rock [26]. Throughout the early vibration process, the strain amplitude increases, and as the vibration further continues, the strain amplitude reaches a maximum and then decreases. Based on Li et al.'s research [27], the natural frequency of rock would change with the rock structure (i.e., deformation and cracks), as shown in Equation (1). Meanwhile, the change in natural frequency would affect the strain amplitude.

$$\omega_n = \sqrt{\frac{aS}{L_y H m}} \quad (1)$$

where ω_n is the natural frequency of the rock sample, Hz; H is the length of the rock sample, m; S is the cross-sectional area of the rock sample, m²; L_y is the length parameter of the crack, m; m is the rock sample mass, kg; and a is the surface energy per unit area, J. Thus, the natural frequency will increase with the decrease in the total length of the cracks (L_y) and the length of the rock sample. The increase in cross-sectional area S will also increase the natural frequency.

Thereafter, during the early process of loading, as the sample compresses and is accompanied by crack closure, both the height of the rock sample and the total length of the cracks decrease, and the cross-sectional area increases. This behaviour would lead to an increase in the natural frequency. As the natural frequency of the rock increases and approaches the ultrasonic vibration frequency, the strain amplitude of the rock would gradually increase. At the same time, the elastic modulus of the rock would first increase under small-stress cyclic loading [9]. Therefore, in the early stage, the stress amplitude increases, and the fatigue process accelerates. After this early stage, closed cracks would reopen and propagate via the generation of new cracks, which would lead to a decrease in the natural frequency and stress amplitude. Thus, to improve the fragmentation efficiency of this technology, the applied ultrasonic vibration frequency should be changed in real time to approach the natural frequency of the rock.

IV. PARTICLE FLOW MODELLING OF GRANITE

A FLAT-JOINT MODEL

PFC, based on the principle of the DEM, can be used to study the evolution of cracks in brittle rocks [18]. Considering the computer performance, we used PFC2D in the present work. In PFC2D, a material is modelled as collections of particles (rigid circular disks). Each particle is in contact with the neighbouring particles via the contact model. The parallel-bond model (PBM) has often been used to simulate crack initiation and propagation in rock or rock-like materials [18, 28-30]. However, there are several issues concerning this contact model when simulating brittle hard rock [31]: (1) an unrealistically low unconfined compressive strength to tensile strength ratio, (2) an excessively low internal friction angle, and (3) a linear strength envelope. To solve these problems, Potyondy [32] proposed a new bond model called the flat-joint model (FJM). Wu Shunchuan, Xu Xueliang and other researchers [31, 33, 34] have indicated that its high reliability for hard rock simulation based on uniaxial compressive strength (UCS) test and Brazilian tensile strength (BTS) test simulations. Thus, in this study, we chose the FJM to perform numerical simulations of granite rocks under ultrasonic vibration.

After a flat-joint contact is established between two particles, forces are generated following the judgement of crack generation. The normal force is computed directly, and the shear force in an incremental fashion [35].

The normal stress $\sigma^{(e)}$ and shear stress $\tau^{(e)}$ acting on an element are given by:

$$\sigma^{(e)} = \frac{-\overline{F_e^n}}{A^{(e)}} \quad (2)$$

$$\tau^{(e)} = \frac{\overline{F_e^s}}{A^{(e)}}$$

where $\overline{F_e^n}$ and $\overline{F_e^s}$ are the normal and shear forces acting on the element and $A^{(e)}$ is the area of the element.

The interface normal stress σ_i is given by:

$$\sigma_i(r) = \begin{cases} 0, & \text{unbonded and } g(r) \geq 0 \\ k_n g(r), & \text{otherwise} \end{cases} \quad (3)$$

where k_n is the normal stiffness and g is the interface gap.

The normal force-update law for a flat-joint contact is as follows:

$$\overline{F_e^n} = \int_e \sigma dA \quad (4)$$

An analytical expression for this integral is used to update $\overline{F_e^n}$, which is then used to update $\sigma^{(e)}$ in Equation (2).

The shear force-update law for a flat-joint contact is as follows:

$$\overline{F_e^s} = k_s A^{(e)} \delta_s \quad (5)$$

where K_s is the tangential stiffness and δ_s is the tangential displacement. Then, this result is used to update $\tau^{(e)}$ in Equation (2).

The tensile strength is specified as σ_b . If the normal stress $\sigma^{(e)} > \sigma_b$, then the element breaks, and a tensile crack generates.

The procedure considering the shear strength based on the bond state is as follows:

(1) Unbonded

The shear strength obeys the Coulomb sliding criterion.

$$\tau_r = \begin{cases} -\sigma^{(e)} \tan \phi_r, & \sigma^{(e)} < 0 \\ 0, & \sigma^{(e)} = 0 \end{cases} \quad (6)$$

where τ_r is the residual friction strength and ϕ_r is the residual friction angle. If $\tau^{(e)} \leq \tau_r$, then the shear stress remains as $\tau^{(e)}$. Otherwise, the shear-strength limit is enforced, and slip occurs.

(2) Bonded

The shear strength follows the Coulomb criterion with a tension cut-off.

$$\tau_c = c_b - \sigma^{(e)} \tan \phi_b \quad (7)$$

where c_b is the bond cohesion and ϕ_b is the local friction angle. If $\tau^{(e)} \leq \tau_c$, the shear stress remains as $\tau^{(e)}$. Otherwise, the shear-strength limit is exceeded. The bond breaks, and a shear crack generates. Then, the residual friction takes effect.

B MICRO-PARAMETER CALIBRATIONS

The microscopic parameters calibrated from macroscopic properties obtained in the mechanical laboratory are considered to be an important procedure for PFC simulation. Only after this process can the established model be used in the next stage.

1) FJM GENERATION

To reproduce the granite sample, the 'clustered assembly approach' was used to generate a randomly distributed-grain granite sample. The model generation procedure is as follows:

Step 1: Two intact numerical specimens are generated first, and the scale of the numerical specimen is 36 mm in diameter and 72 mm in height in the uniaxial compression test simulation and 36 mm in diameter in the Brazilian test simulation. The particle size is controlled by predefined maximum and minimum radii ranging from 0.35 mm to 0.58 mm, considering the computer performance. Each numerical specimen is discretized into 3346 particles with 8513 contacts in the uniaxial compression test simulation and 1360 particles with 3763 contacts in the Brazilian test simulation.

Step 2: To reduce the complexity, it is assumed that there are three minerals in granite, and the contents of these three minerals (i.e., quartz, feldspar and biotite) are 31%, 51% and 18%, respectively. The clustered FJM in the UCS test simulation generates according to the particle extend

method shown in Fig. 8. Each cluster in the model represents a rock grain. As the grain size decreases in the range of albite, quartz and biotite, in the present model, the particle number in each mineral cluster is assumed as follows: feldspar (30 ± 10), quartz (19 ± 8), and biotite (12 ± 5). The first cluster starts to generate from two connected particles (Fig. 8b). It further extends until reaching the required size (Fig. 8c). Then, the second cluster starts to generate in the same way as the first one (Fig. 8c). Through repetition, the generation process stops until the sample is filled with clusters (Fig. 8d). At last, those clusters are grouped according to the mineral component proportions (Fig. 8d), and their density property is assigned according to their actual property (quartz: 2650, feldspar: 2600, and biotite: 2900 kg/m³).

2) CALIBRATION PROCEDURE

After establishing the clustered FJM, we need to import the contact information (i.e., micro-parameters) into the model. There are many micro-properties of the flat-joint contact (e.g., effective moduli, normal/shear stiffness, flat-joint tensile strength, and cohesion) that need to be determined from macro-properties (e.g., Young's modulus, Poisson's ratio, UCS and BTS). Generally, for granite rock, Young's modulus decreases in the order of quartz, feldspar and biotite, and Poisson's ratio decreases in the order of biotite, feldspar and quartz. Similarly, the UCS decreases in the order of quartz, feldspar and biotite, and the BTS decreases in the order of feldspar, quartz and biotite [36, 37].

Apart from the abovementioned characteristics, the collapsing strength for the granular interior of granite is higher than the intergranular strength [30]. Thus, those contacts inside the clusters (intragranular contacts) are assigned a higher bonding strength, while the strength of intergranular contacts, which link different clusters (Fig. 8d), is smaller.

The uniaxial compression and Brazilian tests and corresponding simulations were performed to calibrate the micro-properties. When uniaxial compression test simulations were performed, the loading velocity of the upper and bottom walls is 0.05 m/s, and for the Brazilian test simulations, the loading velocity is 0.0075 m/s, under which the specimen could be regarded in a quasi-static state [33, 38]. The loading wall width in the Brazilian test simulations is equal to two average particle diameters (2.68 mm), and to ensure the smoothness of the specimens, a circular wall with a resolution equal to 0.08 was used to generate samples, as suggested by Xu et al. [33]. The calibration procedure is shown in Fig. 9, and the details are shown as follows:

Step 1: g_{ratio} , φ_s and N_r are first selected aiming to achieve a reasonable coordination number and calculation efficiency [39]. Additionally, as a first approximation, all the other parameters are taken to be the laboratory parameters.

Step 2: Calibration of the UCS and Poisson's ratio: the uniaxial compression test simulations were carried out. The UCS of the model is mainly correlated to c_b , c_b/σ_b and k_n/k_s . The Poisson's ratio of the model is found to be mostly correlated to k_n/k_s [34, 39]. These two macro-properties were matched by iteratively adjusting the micro-parameters until the errors of the macroscopic parameters are within 10%.

Step 3: Calibration of the BTS: Brazilian test simulations were run on the numerical specimen, with the parameters obtained from Step 2, by adjusting σ_b to match the BTS.

Step 4: Calibration of Young's modulus: UCS test simulations were carried out to match Young's modulus at last by adjusting $emod$. The reason for the final calibration is that the micro- $emod$ approximately only affects Young's modulus, while other macro-properties are approximately unaffected [34].

What should be noted is that each time we change values, the micro-properties influence all macro-properties, thus, an iterative calibration is needed after each adjustment.

The micro-parameters obtained after the calibration are listed in Table 1. Furthermore, the calibrated simulated failure modes and macro-properties show excellent agreement with the laboratory test (Table 2 and Fig. 10). Hence, further simulation was based on this rock model.

C CRACK MECHANISMS IN THREE MECHANICAL TESTS

It is widely accepted that tensile cracking is the main damage mechanism of brittle rock in compression tests, such as the Brazilian test and uniaxial compression test. Most of the tensile cracks formed during these two tests are caused by compression [40]. Meanwhile, in the direct tension test, tensile cracks are induced by tension [41].

In this section, the orientation of cracks (the angle between the crack dip and horizontal direction) in the UCS, BTS and direct tension test simulations are obtained. The numerical samples in the tension test simulation are the same as those in the UCS simulation. The loading rate is set as 0.01 m/s. Figure 11 shows the tensile crack orientations from these three tests simulations. In the UCS and BTS tests, although the number of cracks varies greatly, the direction of most tensile cracks is in the range of 60 to 120°, which are parallel and sub-parallel to the loading direction. In the direct tension test, the crack orientation law is different from that in the other two tests, in which the mean orientations of the tensile cracks are in the ranges of 0° to 30° and 150° to 180°, which are perpendicular and sub-perpendicular to the loading direction. The rose diagram of crack orientations shows that the crack orientation obtained by the PFC simulations can effectively reflect the mechanisms of crack evolution.

V. DEM MODELLING OF THE UVBR PROCESS

A CRACK MECHANISMS IN THREE MECHANICAL TESTS

To ensure the pre-requisites determining the success of modelling, four key points should be considered in the UVBR simulations:

(1) The static and dynamic loading conditions should be simulated.

(2) The viscous boundary condition should be applied along the bottom of rock samples to prevent failure due to the reflection of a wave from the boundary.

(3) The attenuation characteristic of ultrasonic vibration propagating in the rock materials should be simulated.

(4) The DEM fatigue damage model is needed to reproduce the gradual damage and hysteresis loops characteristic of rock when subjected to ultrasonic vibration.

The above four points are further detailed in the following parts.

B LOADING CONDITION

Pavlovskaja [42] declared that a low-dimensional model (Fig 12c) can well-simulate ultrasonic vibration systems. In this model, the ultrasonic dynamic loading is applied to the rock object through a block with mass m_2 . The static loading force is directly applied to the block. This model was introduced into our study, and we need to use one kind of element to simulate both static and dynamic loadings.

In general, three common objects can be used to simulate loading conditions in PFC 5.0 (i.e., wall, ball and clump) [30, 35]. The wall object does not obey the equations of motion. Thus, it can be only applied with a velocity, which means it is applicable for the static or quasi-static situation. For the dynamic situation, the ball and clump objects are usually used. They can not only be assigned with a velocity property but also a force condition, as these objects obey the equations of motion. However, when using the ball or clump, to decrease the geometry dispersion effect, increasing the wavelength/diameter is needed, and this will greatly increase the number of balls and pebbles (the pebble is the basic element of the clump), further increasing the calculation time. Furthermore, each pebble of a clump is considered as a piece, often resulting in multiple contacts between a multi-pebble clump and another piece. Such an increase in the number of contacts can also be detrimental to computer performance.

In PFC 6.0, a new object called the rigid block (rblock) is presented [43]. Apart from the basic characteristics of the ball and clump, the rblock can directly model a convex object and does not need to use several particles. Thus, we could use one rblock element to simulate the loading plate, which would greatly decrease the calculation time (Fig. 13). In the present work, we chose the rblock element as the loading plate.

The applied numerical loading strategy is the same as that used in the laboratory tests shown in Fig. 12. A vertical

force (200 N) is assigned to the rblock to simulate static loading. For simulating the dynamic loading, the rblock is assigned a velocity condition, and it is set as follows:

$$v_y = D_{Au} \cdot 2\pi f \cos(2\pi ft) \quad (8)$$

where D_{Au} is the displacement amplitude of the ultrasonic vibration, m ; v_y is the velocity of the rblock in the y direction, m/s ; and t is the loading time, s .

C VISCOUS BOUNDARY

In the modelling of the UVBR process, a viscous boundary should be applied along the bottom of the sample (Fig. 12b) to prevent failure due to the reflection of a wave from the boundary, especially in the high-frequency loading simulation [44]. As there is no built-in viscous boundary in PFC, a possible solution is to release the boundary balls in each timestep via the FISH function [45].

When a wave hits a viscous boundary, a symmetric wave would be generated that could counteract the incoming one. This symmetric wave is applied with minus equivalent stresses as the incoming one:

$$\begin{cases} \sigma_\alpha = -C_p \cdot \rho \cdot \dot{u}_{n-body} \\ \tau_\alpha = -C_s \cdot \rho \cdot \dot{u}_{t-body} \end{cases} \quad (9)$$

where σ_α is the applied normal stress, Pa; τ_α is the applied tangential stress, Pa; ρ , C_p and C_s are the medium's density, kg/m^3 , and P and S wave velocities, m/s ; and \dot{u}_{n-body} and \dot{u}_{t-body} are the velocities of the interior of the model, m/s . To simplify the calculation, C_p and C_s are set to the same value C .

Since Equation (9) is based on the continuum assumption, while in the DEM, the boundaries of the model are not planar. The concept of stress in a discontinuous medium differs from that in a continuous medium. The applied stress should be equivalent to the force. For a regular particle distribution pattern, the equivalent expression between force and stress is:

$$\begin{cases} \sigma_e^n = \sum F_n / D \\ \tau_e^t = \sum F_t / D \end{cases} \quad (10)$$

where σ_e^n and τ_e^t are equivalent normal and tangential stresses, Pa; $\sum F_n$ and $\sum F_t$ are the resultant normal and tangential contact forces of the element, N; and D is the diameter of the element, m .

Combining Equations (9) and (10), and due to the random distribution of particle radii to ensure the optimal absorption of the incoming wave, an inhomogeneous coefficient is introduced [45]:

$$\begin{cases} \sum F_n = \beta_p \cdot D \cdot C \cdot \rho \cdot \dot{u}_{n-body} \\ \sum F_t = \beta_s \cdot D \cdot C \cdot \rho \cdot \dot{u}_{t-body} \end{cases} \quad (11)$$

where β_p and β_s are the normal and tangential inhomogeneous coefficients. To simplify the calculation, β_p and β_s are set to the same value, β .

We monitored the velocities of three balls in the top, medium and bottom parts of the sample to measure the wave velocity and obtain the absorption effects. From the measurement, C is equal to 4300 m/s in the model. The velocities of the three balls with β of 0.4, 0.5, 0.6 and 0.7 are shown in Fig. 14. It can be found from the figure that when β is 0.5, the absorption efficiency is optimum.

D ATTENUATION CHARACTERISTIC OF ULTRASONIC VIBRATION

As shown in Fig. 6, local fragments generate in the UVBR process, and the evident failure occurs on the top of the sample, while the lower part is relatively complete. As an acoustic wave radiates from its source, geometric diffusing reduces the energy per unit area, which is related to the distance from the source. Since the amplitude of the acoustic wave is proportional to the square root of the energy, the pulse amplitude decreases with the distance from the source. Additionally, as the relative loss of energy per cycle is a constant, a pulse with a higher frequency would attenuate more rapidly than a pulse with a lower one, leading to a decent decrease in the pulse amplitude with increasing propagation distance [22]. Thus, it is essential to introduce the attenuation characteristic into the numerical model. One simple and effective way to realize this property is applying a damp property to the particles. The trial-and-error method based on the macroscopic mechanical properties obtained from experiments is used to determine the value of particle damp.

In the experiment, four strain gauges were adhered to the surface of the rock sample in the axial direction with heights of 10 mm, 27 mm, 44 mm and 61 mm. A specific value of the strain amplitude obtained at different heights is 1:0.83:0.7:0.57. This value is used to calibrate the damp parameter in the simulation. After calibration, the particle damp is chosen as 0.5. The numerical specific value of the strain amplitude is 1:0.85:0.73:0.61.

E FATIGUE DAMAGE MODEL

As mentioned in Section 4, the FJM can reproduce the heterogeneity and macro-properties of hard rock. However, it is unable to capture the fatigue damage growth in rock materials. This issue has indeed limited the applicability of PFC for modelling fatigue cracking. To overcome this limitation, a reduction model was proposed by Liu et al. [46] to simulate the fatigue behaviour of an intermittently jointed rock sample under low-frequency uniaxial compression. However, as there are many differences between the low- and high-frequency loading conditions (i.e., the attenuation characteristic would cause significant fatigue anisotropy, and the small-stress cyclic loading would first make rock sample more ‘compacted’), it is not appropriate in the present work. Thus, this model should be updated to make it suitable for ultra-dynamic conditions. In the updated model, the fatigue anisotropy and stiffness characteristics of rock under ultrasonic vibration are

considered. The fatigue damage model methodology is described in detail as follows.

1) MODEL FORMULATION

According to Sadd et al.’s research [47], when rock fatigue occurs, the unloading stress-strain path lies below the loading curve, energy is lost and damage accumulates. Thus, the basic idea behind simulating the rock fatigue is to vary the bond strength and stiffness at different stages (the loading, unloading and reloading stages) [48]. For the ultrasonic vibration condition, rock fatigue is characterized by anisotropy; accordingly, we established the rock fatigue model under ultrasonic vibration as follows:

$$E = \begin{cases} E_0 \times \left(\frac{u_l^i}{u_0^i}\right)^a, \dot{u} > 0 \\ E_0 \times \left(\frac{u_u^i}{u_0^i}\right)^b, \dot{u} < 0 \\ E_0 \times \left(\frac{u_r^i}{u_0^i}\right)^c, \dot{u} > 0 \end{cases} \quad S = \begin{cases} S_0 \times \left(\frac{u_l^i}{u_0^i}\right)^d, \dot{u} > 0 \\ S_0 \times \left(\frac{u_u^i}{u_0^i}\right)^e, \dot{u} < 0 \\ S_0 \times \left(\frac{u_r^i}{u_0^i}\right)^f, \dot{u} > 0 \end{cases} \quad (12)$$

where $\dot{u} > 0$ and $\dot{u} < 0$ denote the loading and unloading stages, respectively; E and S are the effective modulus and bonding strength (i.e., cohesive strength and tensile strength), respectively; and E_0 and S_0 are the initial effective modulus and bonding strength. Considering the fatigue anisotropy, the established model is divided into 200 parts (both the particles and contacts), as shown in Fig. 15. u_0^i is the initial height of the i th part of the rock sample (i is from 1 to 200), and u_l^i , u_u^i and u_r^i are the height of the i th part at the loading, unloading and reloading stages, respectively. To simplify the calculation, the stiffness coefficients (i.e., a , b and c) and strength coefficients (i.e., d , e and f) are set to the same values, which are represented by m and n , respectively. These coefficients will be calibrated according to the experimental results.

2) CYCLE JUMP TECHNIQUE

As seen in Fig. 5, rock fails after millions of cycles under ultrasonic vibration; thus, the UVBR simulations would require extensive computational time if modelling cycle by cycle. To reduce the calculation time, the ‘cyclic-jump’ technique was used. This technique was first presented by [49], and its assumption is that damage steadily develops in a small range of load cycles, which means that during that range the simulation process can ‘jump’ from the first load cycle to the final load cycle. Additionally, the number of cycles omitted in a ‘cycle jump’ is determined according to the following equation:

$$N_{jump} = \frac{\delta D_{jump}}{\delta D} \quad (13)$$

where N_{jump} is the number of omitted cycles, δD_{jump} is an estimated damage increment after a cycle jump, and δD is the average damage increment in the cycle jump range.

The damage increment in the present study is set as the ratio of the residual strain to the maximum residual strain. To simplify the calculation, N_{jump} is set to 500, and in each second, the cycle number can be reduced to 60 cycles (we call them the 'simplified cycles' in the following), which saves considerable computational time. Figure 5 shows that the residual strain rate is approximately $10 \mu\epsilon/s$, which means that the strain would only change $0.16 \mu\epsilon$ per simplified cycle; thus, this simplification is reliable [50].

3) FATIGUE MODEL CALIBRATION

The fatigue damage model was calibrated by adjusting parameters (i.e., m and n) until the numerical strain result matches the experimental one. After calibration, m and n are set as -0.4 and 1.1 , respectively. Figure 16 depicts the dynamic strain curves obtained from the numerical simulations and the experiments. A reasonable consistency can be observed; although in the last stage, compared with the experimental result, the numerical strain value is relatively small. This difference is because the fragments generated in the simulation may not be separated from the matrix, as they actually are in the experiment.

The failure pattern in the numerical simulation is shown in Fig. 17. The irregular blocks with different colours represent fragments, and the azury one is the matrix of the sample. The fragmentation process is similar to that in the experiments, with fragments first occurring at the top corner (in this case in the top-left corner) and then propagating to the adjacent area. In addition, with the increase in cycles, the fragment generation rate increases, which is also consistent with the experimental result.

VI. DYNAMIC MECHANICAL BEHAVIOUR AND THE MICROCRACKING PROCESS

A DISTRIBUTION OF STRESS AND STRAIN FIELDS

One advantage of numerical simulations compared with laboratory tests is that they can directly detect the stress and strain fields. In this section, both the stress and strain fields were obtained based on the measured circle logics shown in Fig. 12b. There are 7×14 circles fully covering the rock sample area.

1) STRESS FIELD

Figure 18 illustrates the distribution of stress under ultrasonic vibration. It can be found that the stress increases with the calculation time, and the stress concentration also becomes evident. First, the stress concentration occurs at the top-left corner, then moves to the centre, and at last to the right. As a result, the whole stress field is divided into three layers and the stress concentration is located at the junction of the first and second layers. This behaviour is because the initial fragments at the top-left corner cause a new significant stress concentration. As fragments continue to emerge, the stress concentration moves accordingly. The development of the stress field corresponds to the generation of fragments.

2) STRAIN FIELD

As the failure process including the damage evolution, crack nucleation and propagation can be well described by the development of a strain field [51], the strain fields during the different loading stages were obtained.

Figure 19 shows the development of the x-direction strain field. As shown in Fig. 19a-b, after a period of loading, the regions of strain concentration are first localized in the middle and upper regions of the specimen. The strain concentration in this range is slight, which may be caused by the mechanical property differences between different minerals. Then, the concentration evolves into a narrower zone (Fig. 19c) and becomes significant. Such a localization of strain reflects the evolution of microcrack damage caused by compression during cyclic loading. Subsequently, the maximum value of local strain increases sharply from 13×10^{-3} to 33×10^{-3} , and the strain at the top-left corner also increases evidently, which indicates that the fragments are generated in this area (Fig. 19d-e). With the further increase in cyclic number, the strain field remains stable (Fig. 19f-h). The y-strain field develops similar to the x-direction one (Fig. 20).

B MICROCRACK EVOLUTION

The acoustic emission (AE) counts and the development of damage and fragments used to declare the cracking process are shown in Fig. 21. The AE counts in the present work represent the generated crack number in each 50 timesteps, as Hunt et al. found that the number of microcracks in the PFC2D is analogous to the AE count [52]. One fragment in the PFC represents one block that contains one or more balls without bonded contacts in its boundary [35]. The mentioned damage is quantified by the following equation:

$$\eta = \frac{\sum_{i=1}^n r_i l_i}{\sum_{i=1}^N r_i l_i} \quad (14)$$

where N and n are the total crack number at ultimate failure and current crack number cumulated, respectively; r_i and l_i are the radius and length of the i th crack. The moment when the calculation is over corresponds to the damage state $\eta = 1$. The initial damage state is $\eta = 0$.

Based on the characterization of AE counts, damage and fragments, the fatigue process of rock under ultrasonic vibration can be divided into 5 stages (Stages 1, 2, 3, 4 and 5). In Stage 1, no cracks occur. From point A, microcracks are generated corresponding to the crack initiation stage (Stage 2). From point B, the cracks develop at a high rate and are followed by the generation of fragments, which corresponds to Stage 3. In Stage 4, the new cracks generate slowly, and a few fragments develop. Point D represents the final failure state, and a large number of fragments are generated during Stage 5.

To further detail the development of cracks in each stage during the UVRB process, local damage was also introduced. The local damage is calculated by Equation

(15), and its distribution is shown in Fig. 22d, which is obtained by dividing the sample area into 10×20 parts:

$$\eta_k = \frac{\sum_{i=1}^{n_k} r_i l_i}{\sum_{i=1}^{N_k} r_i l_i} \quad (15)$$

where N_k and n_k are the total crack number in the most damaged area at ultimate failure and the current crack number cumulated in the k th part of the sample, respectively. The area where the damage is severest at the moment when the calculation is over corresponds to the damage state $\eta_k = 1$. The initial damage state in each part is $\eta_k = 0$. Simultaneously, combining the microcracks' orientation distribution and type in Fig. 22a-c, we can summarize the following:

(1) No-crack stage. In this stage, the tensile x-strain gradually increases; however, it has not reached the crack initiation threshold, so no cracks are generated.

(2) Crack initiation stage. Initial microcracks are distributed at the top-left region, where the tensile x-strain is concentrated. All of the features are tensile cracks that result from compression. These crack orientations are limited to the range $70 < \theta < 110^\circ$, which includes parallel and sub-parallel to the loading direction.

(3) Accelerating crack stage. It can be seen from the AE frequency that the microcracks are generated at a high rate in this stage. The orientations of the tensile cracks are the same as those of the previous stage. Additionally, the cracks gradually propagate to the right area. As some cracks coalesce with each other, the fragments begin to form.

(4) Stable stage. Influenced by the effective depth, the cracks are generated slowly in this stage. The upper part of the whole sample is approximately full of cracks. The effective region is approximately completely destroyed, which means that there is little area to generate new microcracks. Apart from the tensile cracks, a few shear cracks are also generated. Additionally, the tensile cracks' orientation distribution has become more homogeneous.

(5) Final failure stage. With the further increase in strain, new cracks begin to connect with other cracks to generate a shear band, which results in a sharp increase in fragments, and some secondary cracks are generated accordingly, which increase the damage. The significant damage area increases. The shear microcrack orientations are limited to the ranges $0 < \theta < 30^\circ$ and $150 < \theta < 180^\circ$, which correspond to vertical and sub-vertical to the loading direction.

Based on the analysis of the numerical and experimental results described above, the mechanism for the formation of cracks under ultrasonic vibration is revealed. For the tensile cracks (Fig. 23a), as the compressive y-strain and corresponding tensile x-strain increase, two connected particles part from each other in the x-direction, which causes the contact between the two particles (Particle 2 and

3) to experience tensile strain. As the loading continues, the tensile strain increases, and the tensile strain limitation, which controls the contact bond state between particles [53], decreases. When the tensile strain exceeds the tensile strain limitation, the crack perpendicular to the x-axis forms. For the shear cracks (Fig. 23b), as the strain concentration becomes more evident, the strain difference between the strain concentration area and its surrounding area becomes larger. Due to the strain difference, particles (Particle 1 and Particles 2 and 3) at the interface produce a strong dislocation, which finally leads to the generation of shear cracks.

VII. DYNAMIC MECHANICAL BEHAVIOUR AND THE MICROCRACKING PROCESS

In this study, we investigated the fatigue behaviour and crack evolution mechanics of granite under ultrasonic vibration with experimental tests and discrete element simulations. Based on the obtained experimental and numerical results, the following conclusions can be drawn.

(1) Tested intact granite is a typical brittle rock that undergoes violent local failure under ultrasonic vibration. The strain curves first decreased, then remained steady (with small fluctuations), and finally increased. During the first stage, the rock experienced compressive deformation caused by its non-linear property, and no apparent rupture occurred. As the vibration continued, the compressive deformation decreased due to the initiation and propagation of cracks, and finally, significant fragments were generated. The rupture first developed from one point at the edge to the adjacent area. In addition, during the early stage, the stress amplitude of the sample gradually increased due to the increase in the natural frequency and modulus caused by the compressive deformation and corresponding crack closure. The rock fatigue would accelerate due to an increase in the stress amplitude, and this modification would be beneficial to rock breaking. To effectively break hard rock, the applied ultrasonic vibration frequency should be within the range of the varying natural frequency of the rock.

(2) To understand the crack evolution mechanism of brittle granite specimens under ultrasonic vibration, numerical simulations by PFC2D were performed, and an optimal fatigue FJM was proposed, which could not only highly reproduce the properties of hard brittle rock but also capture the fatigue damage characteristic under high-frequency loading. The numerical simulation results showed good agreement with the experimental results in terms of the development of fragments and strain. By analysing the microcrack orientation distribution and crack type, crack evolution was found to include five stages, namely, the undamaged stage, crack initiation stage, accelerated cracking stage, stable stage and final failure stage.

(3) The compression-induced tensile cracks were generated due to the separation of particles resulting from the increase in strain, while the dislocation-induced shear cracks were

generated due to the increase in strain difference between the strain concentration area and its surrounding area.

These experimental and numerical results are expected to improve the understanding of the fatigue damage mechanism of hard rock under ultrasonic vibration and to further increase the crushing efficiency of breaking hard rock.

ACKNOWLEDGEMENTS

This work was supported in part by the National Natural Science Foundation of China under Grant 4157020248, and in part by Graduate Innovation Fund of Jilin University under Grant 101832018C046, which is greatly appreciated.

DATA AVAILABILITY

The fish code used for establishing clustered rock model on networks are publicly available at the following link: ...

Data and relevant code for this research work are stored in GitHub: [\[https://github.com/Python-Zhou/PFC-public.git\]](https://github.com/Python-Zhou/PFC-public.git) and have been archived within the Zenodo repository: <http://doi.org/10.5281/zenodo.3626246>.

CONFLICTS OF INTEREST

The authors declare that they have no conflict of interest.

References

- [1] Q. Geng, Z. Wei, H. Meng, F.J. Macias and A. Bruland, "Free-face-Assisted Rock Breaking Method Based on the Multi-stage Tunnel Boring Machine (TBM) Cutterhead," *Rock Mech. Rock Eng.*, vol. 49, no. 11, pp. 4459-4472, 2016, doi: 10.1007/s00603-016-1053-6.
- [2] L.C. G., K. M., B.C. J. and D.G. I., "A Self-Optimizing Control System for Hard Rock Percussive Drilling," *IEEE/ASME Transactions on Mechatronics*, vol. 13, no. 2, pp. 153-157, 2008, doi: 10.1109/TMECH.2008.918477.
- [3] I. H., V.L. I., A. H. and N. I., "Drilling of hard rocks by pulsed power," *Ieee Electr. Insul. M.*, vol. 16, no. 3, pp. 19-25, 2000, doi: 10.1109/57.845023.
- [4] M. Wiercigroch, J. Wojewoda and A.M. Krivtsov, "Dynamics of ultrasonic percussive drilling of hard rocks," *J. Sound Vib.*, vol. 280, no. 3-5, pp. 739-757, 2005, doi: 10.1016/j.jsv.2003.12.045.
- [5] S. Yin, D. Zhao and G. Zhai, "Investigation into the characteristics of rock damage caused by ultrasonic vibration," *Int. J. Rock Mech. Min.*, vol. 84, pp. 159-164, 2016.
- [6] D. Zhao, S. Zhang, Y. Zhao and M. Wang, "Experimental study on damage characteristics of granite under ultrasonic vibration load based on infrared thermography," *Environ. Earth Sci.*, vol. 78, no. 14, 2019, doi: 10.1007/s12665-019-8450-6.
- [7] Y. Zhou, S. Yin and D. Zhao, "Effect of Static Loading on Rock Fragmentation Efficiency Under Ultrasonic Vibration," *Geotechnical and Geological Engineering*, vol. 37, no. 4, pp. 3497-3505, 2019, doi: 10.1007/s10706-019-00814-3.
- [8] Z. Wang, S. Li, L. Qiao and J. Zhao, "Fatigue Behavior of Granite Subjected to Cyclic Loading Under Triaxial Compression Condition," *Rock Mech. Rock Eng.*, vol. 46, no. 6, pp. 1603-1615, 2013, doi: 10.1007/s00603-013-0387-6.
- [9] A. Hsieh, A.V. Dyskin and P. Dight, "The increase in Young's modulus of rocks under uniaxial compression," *Int. J. Rock Mech. Min.*, vol. 70, pp. 425-434, 2014, doi: 10.1016/j.ijrmms.2014.05.009.
- [10] Y. Liu, F. Dai, P. Fan, N. Xu and L. Dong, "Experimental Investigation of the Influence of Joint Geometric Configurations on the Mechanical Properties of Intermittent Jointed Rock Models Under Cyclic Uniaxial Compression," *Rock Mech. Rock Eng.*, vol. 50, no. 6, pp. 1453-1471, 2017, doi: 10.1007/s00603-017-1190-6.
- [11] S. Yang, N. Zhang, X. Feng, J. Kan, D. Pan and D. Qian, "Experimental Investigation of Sandstone under Cyclic Loading: Damage Assessment Using Ultrasonic Wave Velocities and Changes in Elastic Modulus," *Shock Vib.*, vol. 2018, pp. 1-13, 2018, doi: 10.1155/2018/7845143.
- [12] Z. Jiang, H. Deng, T. Liu, G. Tian and L. Tang, "Study on Microstructural Evolution of Marble Under Cyclic Dynamic Impact Based on NMR," *IEEE Access*, vol. 7, pp. 138043-138055, 2019, doi: 10.1109/ACCESS.2019.2935841.
- [13] Z. Jiang, S. Yu, H. Deng, J. Deng and K. Zhou, "Investigation on Microstructure and Damage of Sandstone Under Cyclic Dynamic Impact," *IEEE Access*, vol. 7, pp. 133145-133158, 2019, doi: 10.1109/ACCESS.2019.2929234.
- [14] X. Hu, C. Du, S. Liu, H. Tan and Z. Liu, "Three-Dimensional Numerical Simulation of Rock Breaking by the Tipped Hob Cutter Based on Explicit Finite Element," *IEEE Access*, vol. 7, pp. 86054-86063, 2019, doi: 10.1109/ACCESS.2019.2925427.
- [15] C. Yang, J. Hu and S. Ma, "Numerical Investigation of Rock Breaking Mechanism With Supercritical Carbon Dioxide Jet by SPH-FEM Approach," *IEEE Access*, vol. 7, pp. 55485-55495, 2019, doi: 10.1109/ACCESS.2019.2913172.
- [16] L. Yang, Y. Jiang, S. Li and B. Li, "Experimental and Numerical Research on 3D Crack Growth in Rocklike Material Subjected to Uniaxial Tension," *J. Geotech. Geoenviron.*, vol. 139, no. 10, pp. 1781-1788, 2013, doi: 10.1061/(ASCE)GT.1943-5606.0000917.
- [17] J.B. Zhu, T. Zhou, Z.Y. Liao, L. Sun, X.B. Li and R. Chen, "Replication of internal defects and investigation of mechanical and fracture behaviour of rock using 3D printing and 3D numerical methods in combination with X-ray computerized tomography," *Int. J. Rock Mech. Min.*, vol. 106, pp. 198-212, 2018, doi: 10.1016/j.ijrmms.2018.04.022.

- [18] X.F. Li, H.B. Li, Y.Q. Liu, Q.C. Zhou and X. Xia, "Numerical simulation of rock fragmentation mechanisms subject to wedge penetration for TBMs," *Tunn. Undergr. Sp. Tech.*, vol. 53, no. 5, pp. 96-108, 2016, doi: 10.1016/j.tust.2015.12.010.
- [19] W. Tian, S. Yang, L. Xie and Z. Wang, "Cracking behavior of three types granite with different grain size containing two non-coplanar fissures under uniaxial compression," *Arch. Civ. Mech. Eng.*, vol. 18, no. 4, pp. 1580-1596, 2018, doi: 10.1016/j.acme.2018.06.001.
- [20] Y. Huang, S. Yang and W. Tian, "Crack coalescence behavior of sandstone specimen containing two pre-existing flaws under different confining pressures," *Theor. Appl. Fract. Mec.*, vol. 99, pp. 118-130, 2019, doi: 10.1016/j.tafmec.2018.11.013.
- [21] A.V. Lebedev, V.V. Bredikhin, I.A. Soustova, A.M. Sutin and K. Kusunose, "Resonant acoustic spectroscopy of microfracture in a Westerly granite sample," *Journal of Geophysical Research-Solid Earth*, vol. 108, no. B10, pp. 1-12, 2003, doi: 10.1029/2002JB002135.
- [22] C.K. McKenzie, G.P. Stacey and M.T. Gladwin, "Ultrasonic characteristics of a rock mass," Elsevier, 1982, pp. 25-30.
- [23] ISRM, "Suggested Methods for Determining the Uniaxial Compressive Strength and Deformability of Rock Materials," *Int. J. Rock Mech. Min. Sci. & Geomech. Abstr.*, 1979.
- [24] Z. Sun, Study on the Effect of Ultrasonic Vibration Frequency on Granite Fracture Law, Jilin University, 2017.
- [25] A. Hsieh, In situ stress reconstruction using rock memory, The University of Western Australia, 2013.
- [26] W. Li, T. Yan, S. Li and X. Zhang, "Rock fragmentation mechanisms and an experimental study of drilling tools during high-frequency harmonic vibration," *Petrol. Sci.*, vol. 10, no. 2, pp. 205-211, 2013, doi: 10.1007/s12182-013-0268-3.
- [27] S. Li, T. Yan, W. Li and F. Bi, "Simulation on Vibration Characteristics of Fractured Rock," *Rock Mech. Rock Eng.*, vol. 49, no. 2, pp. 515-521, 2016, doi: 10.1007/s00603-015-0762-6.
- [28] Y.H. Huang, S.Q. Yang and W.L. Tian, "Cracking process of a granite specimen that contains multiple pre-existing holes under uniaxial compression," *Fatigue Fract. Eng. M.*, vol. 42, no. 6, pp. 1341-1356, 2018, doi: 10.1111/ffe.12990.
- [29] S. Yang, W. Tian, Y. Huang, P.G. Ranjith and Y. Ju, "An Experimental and Numerical Study on Cracking Behavior of Brittle Sandstone Containing Two Non-coplanar Fissures Under Uniaxial Compression," *Rock Mech. Rock Eng.*, vol. 49, no. 4, pp. 1497-1515, 2016, doi: 10.1007/s00603-015-0838-3.
- [30] X.F. Li, Q.B. Zhang, H.B. Li and J. Zhao, "Grain-Based Discrete Element Method (GB-DEM) Modelling of Multi-scale Fracturing in Rocks Under Dynamic Loading," *Rock Mech. Rock Eng.*, 2018, doi: 10.1007/s00603-018-1566-2.
- [31] S. Wu and X. Xu, "A Study of Three Intrinsic Problems of the Classic Discrete Element Method Using Flat-Joint Model," *Rock Mech. Rock Eng.*, vol. 49, no. 5, pp. 1813-1830, 2016, doi: 10.1007/s00603-015-0890-z.
- [32] D.O. Potyondy, A Flat-Jointed Bonded-Particle Material For Hard Rock, *Book A Flat-Jointed Bonded-Particle Material For Hard Rock, Series A Flat-Jointed Bonded-Particle Material For Hard Rock*, ed., Editor ed. American Rock Mechanics Association, 2012, pp.
- [33] X. Xu, S. Wu, Y. Gao and M. Xu, "Effects of Micro-structure and Micro-parameters on Brazilian Tensile Strength Using Flat-Joint Model," *Rock Mech. Rock Eng.*, vol. 49, no. 9, pp. 3575-3595, 2016, doi: 10.1007/s00603-016-1021-1.
- [34] U. Castro-Filgueira, L.R. Alejano, J. Arzúa and D.M. Ivars, "Sensitivity Analysis of the Micro-Parameters Used in a PFC Analysis Towards the Mechanical Properties of Rocks," *Procedia Engineering*, vol. 191, pp. 488-495, 2017, doi: 10.1016/j.proeng.2017.05.208.
- [35] Itasca, PFC2D (Particle Flow Code in 2 Dimensions) manual Version 5.0, *Book PFC2D (Particle Flow Code in 2 Dimensions) manual Version 5.0, Series PFC2D (Particle Flow Code in 2 Dimensions) manual Version 5.0*, ed., Editor ed., 2014, pp.
- [36] H. Lan, C.D. Martin and B. Hu, "Effect of heterogeneity of brittle rock on micromechanical extensile behavior during compression loading," *Journal of Geophysical Research*, vol. 115, no. B1, 2010, doi: 10.1029/2009JB006496.
- [37] M. Villeneuve, Examination of geological influence on machine excavation of highly stressed tunnels in massive hard rock, Queen's University, 2008.
- [38] X. Zhang and L.N.Y. Wong, "Choosing a proper loading rate for bonded-particle model of intact rock," *Int. J. Fracture*, vol. 189, no. 2, pp. 163-179, 2014, doi: 10.1007/s10704-014-9968-y.
- [39] G. Xu, C. He, Z. Chen and D. Wu, "Effects of the micro-structure and micro-parameters on the mechanical behaviour of transversely isotropic rock in Brazilian tests," *Acta Geotech.*, vol. 13, no. 4, pp. 887-910, 2018, doi: 10.1007/s11440-018-0636-7.
- [40] J.M. Kemeny, "A model for non-linear rock deformation under compression due to sub-critical crack growth," *International Journal of Rock Mechanics and Mining Sciences and*

- 1
2
3
4
5
6
7
8
9
10
11
12
13
14
15
16
17
18
19
20
21
22
23
24
25
26
27
28
29
30
31
32
33
34
35
36
37
38
39
40
41
42
43
44
45
46
47
48
49
50
51
52
53
54
55
56
57
58
59
60
- Geomechanics Abstracts*, vol. 28, no. 6, pp. 459-467, 1991, doi: 10.1016/0148-9062(91)91121-7.
- [41] J.J. Liao, M. Yang and H. Hsieh, "Direct tensile behavior of a transversely isotropic rock," *Int. J. Rock Mech. Min.*, vol. 34, no. 5, pp. 837-849, 1997, doi: 10.1016/S1365-1609(96)00065-4.
- [42] E. Pavlovskaya, M. Wiercigroch and C. Grebogi, "Modeling of an impact system with a drift," *Physical review. E, Statistical, nonlinear, and soft matter physics*, vol. 64, no. 5 Pt 2, pp. 56224, 2001.
- [43] Itasca, PFC2D (Particle Flow Code in 2 Dimensions) manual Version 6.0, *Book PFC2D (Particle Flow Code in 2 Dimensions) manual Version 6.0, Series PFC2D (Particle Flow Code in 2 Dimensions) manual Version 6.0, ed., Editor ed., 2018, pp.*
- [44] M.N. Badge and S. Karekal, "Fatigue properties of intact sandstone in pre and post-failure and its implication to vibratory rock cutting," 2015.
- [45] X. Zhou, Q. Sheng and Z. Cui, "Dynamic boundary setting for discrete element method considering the seismic problems of rock masses," *Granul. Matter*, vol. 21, no. 3, 2019, doi: 10.1007/s10035-019-0918-2.
- [46] Y. Liu, F. Dai, T. Zhao and N. Xu, "Numerical Investigation of the Dynamic Properties of Intermittent Jointed Rock Models Subjected to Cyclic Uniaxial Compression," *Rock Mech. Rock Eng.*, vol. 50, no. 1, pp. 89-112, 2017, doi: 10.1007/s00603-016-1085-y.
- [47] M.H. Sadd, Q. Tai and A. Shukla, "Contact law effects on wave propagation in particulate materials using distinct element modeling," *Int. J. Nonlin. Mech.*, vol. 28, no. 2, pp. 251-265, 1993, doi: 10.1016/0020-7462(93)90061-O.
- [48] B. Peng, *Discrete Element Method (DEM) Contact Models Applied to Pavement Simulation*, Virginia Polytechnic Institute and State University, 2014.
- [49] J.L. Chaboche, "Continuum Damage Mechanics: Part II—Damage Growth, Crack Initiation, and Crack Growth," *Journal of Applied Mechanics*, vol. 55, no. 1, pp. 65-72, 1988.
- [50] N.H.T. Nguyen, H.H. Bui, J. Kodikara, S. Arooran and F. Darve, "A discrete element modelling approach for fatigue damage growth in cemented materials," *Int. J. Plasticity*, vol. 112, pp. 68-88, 2019, doi: 10.1016/j.ijplas.2018.08.007.
- [51] J.Q. Xiao, D.X. Ding, G. Xu and F.L. Jiang, "Inverted S-shaped model for nonlinear fatigue damage of rock," *Int. J. Rock Mech. Min.*, vol. 46, no. 3, pp. 643-648, 2009, doi: 10.1016/j.ijrmms.2008.11.002.
- [52] S.P. Hunt, A.G. Meyers and V. Louchnikov, "Modelling the Kaiser effect and deformation rate analysis in sandstone using the discrete element method," *Comput. Geotech.*, vol. 30, no. 7, pp. 611-621, 2003, doi: 10.1016/S0266-352X(03)00061-2.
- [53] Y.T. Feng, K. Han, D.R.J. Owen and J. Loughran, "On upscaling of discrete element models: similarity principles," *Eng. Computation.*, vol. 26, no. 6, pp. 599-609, 2009, doi: 10.1108/02644400910975405.

Fig. 1

FIGURE. 1 X-ray diffraction showing mineral compositions of granite.

Fig. 2

FIGURE. 2 Polarized photomicrographs showing the grain size of the granite

Fig. 3

FIGURE. 3 Natural frequency of rock sample

Fig. 4

FIGURE. 4 Ultrasonic vibration strain test: a Ultrasonic vibration and strain testing equipment; b Loading method

Fig. 5

FIGURE. 5 Strain data and failure pattern of 6 samples in vibration strain test (the negative strain represents the contraction of sample and the positive strain represents the dilation of sample)

Fig. 6

FIGURE. 6 Fracture mode of the samples (Area enclosed by red line represent the destruction zone and the arrow points in the destruction direction.)

Fig. 7

FIGURE. 7 Change process of strain amplitude: a Partial enlargement map of strain; b Strain and strain amplitude curves.

Fig. 8

FIGURE. 8 Particles extend cluster method: a the initial intact sample b the beginning of the first cluster (red circles) c the first cluster finishes and the second one starts to generate (green circle) d the sample with all clusters (the black lines represent intergranular contacts and the yellow lines represent intragranular contacts) e the distribution of minerals (light grey cluster represents quartz, dark grey feldspar and black biotite)

Fig. 9

FIGURE. 9 Flowchart of calibration process for flat-jointed models

Fig. 10

FIGURE. 10 Comparison of failure modes obtained by laboratory tests and numerical simulations: a Uniaxial compression test; b Brazilian test (the black line represents shear crack, while the red line is tensile crack).

Fig. 11

FIGURE. 11 Distribution of crack plane orientation for tensile cracks in different simulation tests: a UCS; b Brazilian test; c Direct tension test (ten numerical specimens with different bond strength distributions).

Fig. 12

FIGURE. 12 Loading method in the PFC simulations: a Schematic diagram of experimental device; b Schematic diagram of loading condition and layout of measurement circles (the yellow cycles are measurement cycles and some of them are concealed); c Physical model of the UVBR system (A_u is the velocity amplitude).

Fig. 13

FIGURE. 13 Numerical loading plate: a loading plate consists of many balls or pebbles in PFC 5.0; b loading plate consists of one rblock in PFC 6.0.

FIGURE. 14 Velocity of three monitored balls with different inhomogeneous coefficients β : a 0.4; b 0.5; c 0.6; d 0.7.

Fig. 14

Fig. 15

FIGURE. 15 Different parts of the model: a Particle groups; b Contact groups.

Fig .16

FIGURE. 16 Experimental and simulated strain results (The simulation data is the average of the strain obtained from the measurement circles with id of 1, 8, 15 and 22 whose positions correspond to the strain gauge in the experiment.)

Fig. 17

FIGURE. 17 Development of the fragments: a 0s; b 45s; c 62s; d 95s e 127s

Fig. 18

FIGURE. 18 The distribution of stress in numerical sample under the ultrasonic vibration (a-h represent 4-120s)

Fig. 19

FIGURE. 19 The distribution of x-strain in numerical sample (the negative value represents compressive strain, and the positive value represents tensile strain (a-h represent 4-120 s))

Fig. 20

FIGURE. 20 The distribution of y-strain in numerical sample (the negative value represents compressive strain, and the positive value represents tensile strain (a-h represent 4-120 s))

Fig. 21

FIGURE. 21 Various variables, acoustic emission counts (black bars), damage (red line) and volume of fragments (blue line) and changes as a function of vibration time

Fig. 22

FIGURE. 22 Fracture initiation and propagation and damage evolution process during UVBR process: a DEM simulated cracks' distribution and propagation; b-c Corresponding crack orientation; d Corresponding damage field. Different columns represent different stages 2, 3, 4, and 5 plotted in Fig. 20. Green lines represent tensile cracks and red shear cracks.

Fig. 23

FIGURE. 23 Crack mechanism of rock under ultrasonic vibration: a Compression-induced tensile crack; b Dislocation-induced shear crack

TABLE I
MICRO-PROPERTIES OF BALLS AND CONTACTS

	Quartz	Feldspar	Biotite
General			
Volume composite, V_{ratio}	30.7%	50.6%	18.7%
Particle			
Minimum ball radius, r_{min} (mm)		0.5	
Maximum/Minimum, r_{max}/r_{min}		1.66	
Ball density(kg/m ³)	2650	2600	2900
Intragranular Contacts			
Bonded element fraction, ϕ_B		1	
Slit element fraction, ϕ_T		0	
Total number of elements, N_r		2	
Stiffness ratio	2.4	2.6	3.0
Effective moduli (GPa)	78	50	30
Mean and standard deviation bond tensile strength, σ_b (Mpa)	18±1.8	27.5±2.75	12.5±1.25
Mean and standard deviation bond cohesion, C_b (Mpa)	100±10	88±8.8	60±6
Local friction angle, ϕ_b (°)	3	5	8
Residual friction angle, ϕ_r (°)	7	10	13
Intergranular Contacts			
Stiffness ratio		3.1	
Effective moduli (GPa)		23	
Mean and standard deviation bond tensile strength, σ_b (Mpa)		9.5±0.95	
Mean and standard deviation bond cohesion, C_b (Mpa)		40±4	
Local friction angle, ϕ_b (°)		10	
Residual friction angle, ϕ_r (°)		15	

TABLE II
GRANITE PROPERTIES FROM EXPERIMENTS AND SIMULATIONS

Property	Experiment results	Simulation results
Young's modulus E (GPa)	52.3±12.2(n=10)	52.7±0.26(n=10)
Poisson's ratio m	0.26±0.05(n=10)	0.255±0.006(n=10)
Uniaxial compressive strength (UCS) (MPa)	116±19(n=10)	116.3±3.1(n=10)
Brazilian tensile strength (BTS) (MPa)	8.1±1.5(n=10)	8.1±0.17(n=10)

Note that, in the experiments, n denotes the number of samples undergone the compression or Brazilian tests, while in PFC, it denotes the number of samples with different bond strength distributions

Appendix B

Reviewers' Comments to Author:

Reviewer: 1

Comments to the Author(s)

1) The introduction part needs to be well organized again, and it should focus on the content related to this study.

Thank you for your suggestion. The introduction has been carefully organized again.

2) One dedicated paragraph must be there for explaining the research gap. The motivation for the work should spontaneously come out from the research gap itself which is not evident.

We have rewrite one paragraph (the third paragraph) to explain the research gap. Thanks.

3) It is necessary to give the complete stress-strain curve of the sample and explain why the static load adopted is 200N (Whether 200N reaches the fatigue triggering condition).

Limited by the apparatus, there is no stress data of the sample now. However, the axial strain data is one important data to reflect the fatigue behavior and meaningful to analyze the subject.

We are looking forward to getting the complete stress-strain curve of rock under ultrasonic vibration. The moment when the apparatus could reach this goal, we would conduct some experiments as soon as possible. We think this is meaningful and interesting.

The reason why we choose 200N as the static loading has been mentioned in the manuscript (Section III), that is under this loading parameter the crushing efficiency is higher based on the previous experiments.

4) “Thus, to improve the fragmentation efficiency of this technology, the applied ultrasonic vibration frequency should be changed in real time to approach the natural frequency of the rock.” Is resonance the essence of this sentence?

Yes, the essence is reaching the resonance condition.

5) “A specific value of the strain amplitude obtained at different heights is 1:0.83:0.7:0.57.” No corresponding strain gauge is found in the experimental picture. What is the relationship between strain and damping? How to deal with it?

Thank you for your comment. In this section, the strain amplitude at different heights is used to calibrate the damp factor to simulate the attenuation characteristic of ultrasonic vibration in the numerical simulation. Thus, we have not laid relevant picture there. The schematic diagram it as follows. We have rewritten this sentence to avoid confusion.

6) PFC numerical simulation modeling process should be simplified, part of which is the common sense of software users.

Thank you for your suggestion. We have simplified common parts of our simulation description (i.e. the calibration process and introduction of the mechanism of the viscous boundary in the DEM).

7) How to verify the correctness of numerical simulation results? Please give the stress-strain curve of the simulation result and compare it with the stress-strain curve of the sample.

As mentioned above, we would further contact with the instrument factory to update the apparatus to measure the stress data under ultra-high frequency loading condition. If it would be realized in the further, we would conduct some experiments as soon as possible. We would like to keep in touch with you to discuss this topic. Thanks.

8) Please rearrange the conclusions of this paper and emphasize the innovative work.

Thank you for your suggestion. The conclusion has been rearranged and the innovative contents have been emphasized.

Reviewer: 2

Comments to the Author(s)

1. This manuscript presents new data from a suite of uniaxial compression and Brazilian disc experiments that utilize an ultrasonic vibration device to apply ultra-high frequency loads to the samples of fine grained granite. Some of the experimental data was used to calibrate a suite of PFC numerical simulations which were in turn used to explain observations from the ultra-high-frequency loading tests. The numerical simulations incorporate a new flat-joint method of coupling bonds which reduces or eliminates issues related to the more commonly used parallel-bond method. The experimental results demonstrate three main phases of deformation resulting from the ultrasonic vibrations, discerned from a single strain gauge positioned at the upper most part of the sample. The numerical results are more numerous and this seems the main focus of the paper in reality. Resultant stress and strain fields are reproduced in the models and Acoustic Emission output is also considered as a proxy for crack damage. It seems strange to me that the authors did not monitor Acoustic Emission output in the experiments as this would provide an even more robust check of the numerical data and especially permit confirmation of fatigue or damage memory effects. In all I found the descriptions of the experimental methods and results very difficult to follow and I suggest these parts require substantial re-writing. I also thought that the Introduction did an inadequate job of explaining the motivation for the study and the section felt quite disjointed the way it has been written. Again, I suggest this section be reworked.

Due to the lack of apparatus, we have not conducted the AE experiments.

Thank you for your suggestions, we have reorganized the introduction part and the experimental part.

The detailed explanation and discussion can be seen in the response to the comments for the PDF file in the next part.

2. As regards the scientific content, I found the study to be interesting although I have some concerns regarding the experimental setup. To start, what was the motivation for using just one single strain gauge? It is not explicitly clear in the manuscript which component of strain is plotted and discussed throughout. I am concerned that by placing the strain gauge near the upper sample surface, and not recording the strain in the central part of the sample, some of the variety of processes occurring are being obscured. This may be problematic when also considering that the upper and lower surfaces of materials loaded in uniaxial conditions normally experience edge effects. This process is explained in detail in section 3.5 of the book 'Rock deformation the Brittle Field' by Paterson. It seems from the experiments and models that most of the fracture damage is confined to the upper most portion of the rock sample. Was any material placed between the piston and the sample to eliminate such effects?

The samples were carefully cut and polished according to ISRM(UCS) standard. Thus, the edge effect should not be so much at this situation.

Besides, according to the strain amplitude measurement test, the amplitude gradually decrease with the height. Thus, the fragment process that the evident failure occurs on the top of the sample and the lower part is relatively complete is mainly due to the characteristic of ultrasonic vibration.

Thus, we thought the top part of the rock sample is the most important part we should investigate and adhered one single strain gauge there to measure the axial strain. The reason why we measure the axial strain (in the experiment it is the axial residual strain) is that this data could be a good option to describe the fatigue behavior of rock.

3. Please try to avoid the use of qualitative terms such as ‘small’ and ‘large’ for describing the results or cycle numbers, for example. Such terminology can be avoided because the actual values can be given. I have made many comments on a marked up PDF version of the manuscript which I attach. Some of the more urgent comments that I briefly note here relate to Figure 5. Can the authors comment on why the amounts of strain and duration (strain rate) of the experiments are so variable? There is little comment in the manuscript about repeatability and checking this plot leads me to question how repeatable the tests are. On this point, I think a proper characterisation of the material used would be very welcome. There is some information about the grain size but not an examination of the distribution of pre-existing fractures. How heterogeneous or anisotropic is the granite? Where was it collected from etc?

Thanks for your suggestions. We have responded your comments point-by-point and they are listed at the end of the response.

We have added the characteristics of the used granite materials we know in the manuscript.

The rocks are typical heterogeneous and anisotropic materials, which will certainly make the testing results unreproducible. The other factors, e.g. the preparation of samples, the experimental set-up, and environmental effects such as temperature and humidity will also contribute to the discreteness of the testing results. This needs a plenty of tests in a series of future studies, where the samples made from other brittle and relatively homogeneous materials, such as concrete and gypsum, will be adopted.

4. Also, I’m not convinced that crack length is a suitable parameter to use in equation 1. Is it not crack opening or crack surface area that will control the frequency more than simply length?

As this equation has been proposed in the published article, we unconsciously think it’s right. Under your guidance, we have rethought this equation. If just considering the length of crack may be not sufficient. As could be found from Li et al.’s research, the longer or the wider the crack is, the smaller the harmonic frequency of rock is. Thus, the crack surface area should be considered while not just the crack length. In the other hand, as the system natural frequency is directly related to the stiffness, it means that the crack opening width would also influence the frequency. We would like to do some theoretical research to address this problem in the future.

Thus, it may be not reasonable to use previous equation in our present work. In order to interrupting the experimental phenomenon, we have decided to use the basic natural frequency equation as follows. And we have revised all content related to this equation in the manuscript. Thanks.

$$\omega_n = \sqrt{k/m}$$

Overall I feel that the manuscript requires major revision and substantial rewriting before it is reconsidered for publication in Royal Society Open Science.

The comments in the PDF file of the Reviewer2:

1. Abstract → Clarify the 'strain curve first decrease' in the abstract.

Better to discuss this in terms of 'strain' rather than the shape of the 'strain curve'.

The revision is as follows.

The experimental results show that **the strain curve is U-shaped, and it can be divided into 3 stages:** the strain first decreases, then remains steady (with light fluctuations), and finally increases.

2. Abstract → Does this mean that during the first stage of compression there was no rupture? Would rupture be expected during this stage?

The rupture here means macro-rupture. Thus, according to experiments, there is no rupture during this stage.

3. Abstract → What is 'significant' in this case?

We would like to state that there are fragments generated. To avoid confusion, we have removed this adjective. Thanks.

4. Paragraph 1 Section I → Rewrite - incorrect grammar.

Thank you for your suggestion. The revision is as follows.

Due to the fact that the hard and brittle rocks with high strength is widely encountered in underground projects such as tunnelling, petroleum and geological drilling, (e.g. granite), rapidly breaking of hard rocks, is a widespread issue that needs to be optimized[1-3].

5. Paragraph 1 Section I → This is a bit confusing because the sentence starts with a result from the study of Yin et al, but then the final part of the sentence simply states something that they examined, which is not related to first finding. I suggest this be rewritten.

Thank you for your suggestion. The revision is as follows:

The results demonstrated that this technique can effectively reduce the strength of hard granite **when the static loading exceeded the static pressure threshold for this technology as the fatigue damage occurred.**

6. Paragraph 1 Section I → State the static pressure threshold. Is that rock type dependent?

Thank you for your suggestion. We have stated the static pressure threshold in the manuscript.

Based on the existence experiments, as the rock material are with similar characteristics, the static pressure threshold for them are similar. However, we think it should be related to rock

characteristics especially the tensile strength and would like to do some experiments to verify it. Thanks.

7. Paragraph 1 Section I → What is meant by 'rock area'? Do the authors mean this as oppose to the 'soil area' for example?

We meant that the rock sample body can be divided into three zones under ultrasonic vibration and we have revised it in the manuscript to avoid confusion. Thanks

8. Paragraph 1 Section I → What is 'the system' here? The 'rock area'?

We meant the rock sample here and we have revised it in the manuscript to avoid confusion. Thanks.

9. Paragraph 1 Section I → It would be good to state which rocks these other papers studied if not granite.

Thank you for your suggestions and we have stated the used rock type in the cited researches.

10. Paragraph 2 Section I → But this will occur at increasingly higher levels of stress, i.e. the Kaiser 'stress memory' (Holcomb, 1993; Lockner, 1993; Lavrov, 2001) or 'damage memory' (Browning et al., 2017; 2018) effect.

Thank you. I looked at the literature you suggested. I agree with your opinion.

11. Paragraph 2 Section I → Do you mean in the specific case of cyclic triaxial loading or in general? Dissipation of energy in rocks has been studied more thoroughly than is suggested simply by referencing Liu et al.

In general case.

We have cited more relevant literature. Thanks.

12. Paragraph 2 Section I → Can add a line here to state why that is the case, i.e. rather than simply reporting the result indicate the underlying mechanics.

Thank you for your suggestion. We have revised it as follows.

Yang et al. [7] used **the P and S-wave velocities based on** the ultrasonic wave velocity method **to reflect the sandstone behavior (i.e. elastic modulus and the shear modulus) under cyclic loading condition** and found that rock specimens stiffen during loading and soften during unloading.

13. Paragraph 2 Section I → What is the relevance of 'dynamic impact' here?

It is the cyclic loading and we have revised it. Thanks.

14. Paragraph 2 Section I → Are the 'damage variables' mentioned here the same as the porosity and pore size mentioned in the previous sentence?

The damage variable here is a parameter related to the porosity.

$$D = \frac{\phi_i - \phi_{i-1}}{\phi_6 - \phi_0}$$

where subscript i represents the numbers of dynamic impact that samples underwent, which ranges from integer 1 to 6, ϕ_0 and ϕ_6 represent porosity of rock sample at initial state and under 6 dynamic impacts respectively.

15. Paragraph 3 Section I → What do you mean by 'dynamic experiments' in this case? Researchers have used AE (Lockner, 1993) and Ultrasonic wave velocities (Ayling et al., 1995) to monitor crack growth development for many years without great difficulty.

Thank you for your comment. We have deleted this sentence as this is not rigorous.

16. Paragraph 3 Section I → Why is a result reported for study 16 but not studies 14 and 15?

Thank you for your suggestion and we have added relevant results for references 14 and 15 as follows.

Hu et al. [5] used LS-DYNA to study the rock fragmentation mechanisms and concluded that there are three fracture modes of rock breaking: forward slip, no slip, and backward slip. Yang et al. [6] investigated the rock breaking mechanism under supercritical (SC)-CO₂ jet impacting based on the smoothed particle hydrodynamic finite element method (SPH-FEM). Numerical results showed that there were two failure forms for rock under SC-CO₂ jet impacting presented: erosion and exfoliation.

17. Paragraph 3 Section I → Do you mean 'more difficult' or simply that shear cracks are less abundant in coarse grained granites rather than fine grained granites? This is presumably simply because there are less grain boundary contacts in coarse grained materials.

I agree with your opinion. Your comments were added in the manuscript. Thanks.

18. Paragraph 3 Section I → Incorrect grammar.

The revision is as follows, thanks.

To enhance the understanding of failure mechanism of rock, Huang et al. [21] carried out conventional triaxial compression tests simulation by using particle flow code in three dimensions (PFC3D).

19. Paragraph 3 Section I → I presume the authors mean 'sample lengths'. This is of course very well known, see for example Paterson and Wong, 2005 and the manuscripts reference 23.

Thank you for pointing out this mistake.

20. Paragraph 1 Section II. A → Any particular type of granite? Do the authors mean 'fine grained'?

The used granite is general granite.

Yes, it is one kind of fine-grained granite and we have revised it in the manuscript. Thanks.

21. Paragraph 1 Section II. A → Rephrase ‘According to the International Society of Rock Mechanics [23] standards, the sample size was assumed to be large enough to ignore its size effect.’

Thank you for your suggestion. The revision is as follows:

According to the ISRM standard regarding uniaxial compression tests in the laboratory, the slenderness ratio (the ratio of the diameter to the height) of the sample is suggested to be in the range of 2.0 to 2.5. Therefore, the sample prepared in our work obeys the ISRM standard.

22. Paragraph 1 Section II. A → Can the authors comment on how heterogeneous or anisotropic the rocks are? Were they cut from the same block? What was the outcrop condition, any faults, alteration or joints etc?

Did the authors characterize the material prior to testing, i.e. checking the virgin materials porosity or Vp/Vs?

We do not have relevant apparatus to analyze its heterogeneity and anisotropy.

As the topic of the research is to improve hard rock breaking efficiency, we do not need to choose specific rock. Considering the cost, we bought rock blocks from one stone pit in Jilin Province, China. We do not know the outcrop condition. We cut them from a set of rock block. The porosity of the rock sample is about 0.428%. Thanks.

23. Paragraph 1 Section II. B → Give the range of static loads applied here.

The used loading parameters has been listed in the next paragraph. The frequency is 30Khz and the static loading force is 200N.

24. Paragraph 1 Section II. B → This is confusing.

The effective area of the strain gauge is shown as below. As it is easy to confuse reader, we have removed it. Thanks.

25. Paragraph 1 Section II. B → Which components of strain were measured?

The axial component. We have added it the manuscript.

26. Paragraph 2 Section II. B → This should be written in past tense.

Did the authors perform tests with only one static load and frequency?

Thanks. We have revised this mistake.

Yes, we have just performed tests with the frequency of 30 KHz and static loading of 200N in the present work, as the crushing efficiency is higher compared with other parameter combinations based on previous experiments.

27. Paragraph 2 Section II. B → 'Fragmentation' perhaps.

Thank you for pointing out this mistake and we have checked the whole manuscript.

28. Paragraph 1 Section III. → Can you comment on why the absolute amounts of strain are so different between tests? See my comments made directly on Figure 5.

The rocks are typical heterogeneous and anisotropic materials, which will certainly make the testing results unreproducible. The other factors, e.g. the preparation of samples, the experimental set-up, and environmental effects such as temperature and humidity will also contribute to the discreteness of the testing results. This needs a plenty of tests in a series of future studies, where the samples made from other brittle and relatively homogeneous materials, such as concrete and gypsum, will be adopted.

Besides from this, it can be found from the figure, although there are some differences, the maximum compressive strains of these six samples are almost same. This is consistent with previous research [7], and they concluded that the axial residual strain could be a better option to describe the fatigue behavior of rock than the loading cycle number (i.e. loading time).

29. Paragraph 1 Section III. → But it isn't clear how heterogeneous or anisotropic the rocks are. This should be better described in the materials section.

Thank you for your suggestion. However, we do not have relevant apparatus to analyze its heterogeneous and anisotropic.

30. Paragraph 1 Section III. → Give the components of strain or say volumetric strain.

Thanks. We have revised this mistake as follows.

The **axial** strain curve is U-shaped, and it can be divided into 3 stages:

31. Paragraph 1 Section III. → Revise 'gradual' into 'gradually'.

Thank you for your suggestion.

32. Paragraph 1 Section III. → ‘Did it?’ and ‘The sense of these paragraphs is odd. I'm not sure if the authors are describing the results or interpreting something that 'would/could or might' happen.’

We would like to interpret the rock behavior and there are some logic problems here. The revision is as follows. Thank you for you pointing out this problem.

In the first stage, **as the surface of the open crack might not be smooth, the sharp points might break when subject to ultrasonic vibration. The axial stain reduces because of the generation of breakage accompanied by the crack closure. Some energy is consumed in this process and irreversible compressive deformation gradually increases.** Thus, the sample becomes more ‘compacted’. In other words, the modulus of rock would increase.

As the vibration continues, **due to the highly concentrated stress, new cracks gradually form, and the initial cracks would reopen and propagate. Thus, the slope of the strain curve decreases.** These cracking behaviours would result in a decrease in rock modulus and cause rock expansion in the axial direction. In the last stage, as the cracks continue to propagate and coalesce, the magnitude of the expansion would become larger, and the strain increases.

33. Paragraph 2 Section III. → How many samples were tested in total?

Sorry for our leaving the number of samples. Our test mainly contains two parts, one is the strain test and another is fragment observing test. Each set contains 6 samples. Why we conduct fragment test is that the strain gauge would fail when major fragments first occurs. We have added these numbers in the manuscript.

As the stain gauges would fail when large fragment first occurs, to observe the complete fragmentation process of the rock sample under ultrasonic vibration, we conducted the fragment monitoring test and it contains 6 samples. One general case is shown in Fig. 6.

34. Paragraph 2 Section III. → What is the definition of 'influential range'?

This sentence has been rewritten to improve its readability. We have replaced the ‘influential range’ into ‘effective region’ and labeled the effective region in the figure 6. Thanks.

According to the fragment pattern, it can be concluded that **there exists an effective region for this technique among which significant damage generated.**

35. Paragraph 3 Section III. → Is this the max and min of certain components of strain or volumetric?

This is the max and min axial strain.

36. Paragraph 3 Section III. → Why each second? Is this the recording rate?

The record rate is high (10KHz), which means that the variation is little if we calculate the strain amplitude cycle by cycle. Thus, in order to analyze its development law effectively, we calculated it by subtracting the min strain from max strain in each second.

37. Paragraph 3 Section III. → This is confusing, does the elastic moduli affect fatigue or does fatigue change the elastic moduli? From the introduction it would seem that some critical stress level is needed to induce fatigue and hence alter the Elastic Moduli.

We have revised this sentence to make it clear.

The strain amplitude was obtained and combined with the development of the elastic modulus to track the stress amplitude, **and the stress amplitude** can greatly affect the fatigue behaviour of rock **sample**.

38. Paragraph 3 Section III. → This doesn't seem suitable for the results section, i suggest this be moved to the introduction.

Then it should be explained how each of these parameters are measured or interpreted in the methods section.

As mentioned in the Response 4, we have decided to use the basic natural frequency equation. And we have revised all content related to this equation in the manuscript. Thanks.

39. Paragraph 4 Section III. → Does the length of the cracks decrease or the cross sectional area of cracks decrease? Doesn't this depend on the orientation of pre-existing cracks?

I think that this depends on the orientation of the cracks with respect to the loading direction, but i do not think that crack length is an appropriate parameter here as crack length won't change under such circumstances. Cracks aligned with their a axis parallel to the orientation of loading would presumably keep the same length (until a critical stress was reached and they grow) but their c axis would increase in width. Whereas cracks with their a axis aligned normal to the loading direction would close, i.e. the c axis would become less wide, but again, i would expect the a axis to remain constant (until a critical stress is reached).

I agree with your opinion regarding the crack deformation when the initial cracks are parallel to the loading direction. However, the initial cracks in rocks are of different inclination angles and shapes. Therefore, in general, at the initial loading stage, the initial cracks subjected to the loading is experiencing the closure of different extent. The crack growth from the initial crack with the direction of the loading is quite interesting topic which may be the source of the crack growth at the initial loading stage when subject to ultrasonic vibration.

40. Paragraph 4 Section III. → Is this the cross-sectional area of the cracks or the sample?

If the sample, presumably this is related to the Poisson's effect, but what are the implications of measuring this at the sample edge, where the strain gauge is placed? In Paterson, section

3.5 'Rock deformation the brittle field' there is a section on edge effects, it seems that many of the interesting processes that are observed in your experiments occur within Paterson's stress shadow or edge effected zone. Can you comment on that please?

First, the samples were carefully cut and polished according to ISRM(UCS) standard. Thus, the edge effect should not be so much at this situation. As the rock elements near the sample ends is of interest, the strain gauge is attached to the sample ends. Before fracturing of this part, the strain of this part is of key importance to analyze the mechanisms of fracturing of the rock subjected to ultrasonic vibration.

41. Paragraph 4 Section III. → The use of the words 'would and should' in these sections make this all very confusing. I'm not sure what really are the results of these experiments.

Sorry for our statement which confused you. We have revised this section as follows.

In the early stage, due to the crack closure, the stiffness of rock increases and further causing the increase of the natural frequency. The change in natural frequency also affects the strain amplitude as the rock is reaching the resonance condition [34], that is the strain amplitude increases. Thereafter, during the early process of loading, as the sample compresses and is accompanied by crack closure, both the strain amplitude and stiffness of the rock gradually increase. Thus, the stress amplitude increases in this stage, and the fatigue process accelerates. After this early stage, closed cracks will reopen and propagate accompanied by the initiation of new cracks, which will lead to a decrease in the stiffness and natural frequency of rock, and further decrease the strain amplitude and stress amplitude. Thus, to improve the fragmentation efficiency of this technology, the applied ultrasonic vibration frequency should be changed in real time to approach the natural frequency of the rock.

42. Paragraph 1 Section IV. A → What was the computer performance?

The computer performance is related to the computational speed and it is mainly influenced by the CPU, memory and so on.

43. Paragraph 1 Section IV. A → Is this a function of the PFC software or something that was added or coded into the software by the authors?

The flat joint model is a built-in function of PFC software. I have updated this model to make it suitable to simulate ultrasonic vibration process.

44. Paragraph 1 Section IV. C → Citation would be good here.

Thank you for your suggestion.

45. Paragraph 1 Section IV. C → This statement is confusing, i suggest this be rephrased. Whilst the samples in Brazilian tests are loaded in compression, the geometry with respect to the loading direction, locally generates tension hence why the sample splits apart. So i do not think that it is correct to say that the cracks are caused by compression, as they aren't.

Thank you for pointing out our mistakes. We have revised this sentence as follows under your guidance.

In the UCS test, most of the tensile cracks formed are caused by compression. The tip of the pre-existing cracks endures tension under compression and induces tensile wing cracks. Whilst the samples in Brazilian tests locally generates tension force and hence it will split apart.

46. Paragraph 2 Section IV. C → But don't they have different geometries?

The geometries of the samples in the UCS and direct tension are the same.

Figure (a) The sample used in the Brazilian test simulations; (b) The samples used in the UCS and direct tension simulations.

47. Paragraph 2 Section IV. C → This seems contradictory to the statement in the previous section where the loading velocity for the UCS test is 0.05 m/s and the Brazilian tests was 0.0075 m/s.

The principle of choosing loading rate is guaranteeing the specimen could be regarded in a quasi-static state, thus, the loading rate in different test simulation is chosen as different value based on previous literatures.

48. Paragraph 2 Section IV. C → Clarify that you mean the a axis of cracks, i.e the cracks grow in the orientations parallel to the loading direction, but open in orientations normal to the loading direction.

Thank you for your suggestion. We have clarified the direction of cracks based on your guidance and illustrate it in the Figure 9.

49. Paragraph 1 Section V. A → These four points are written as though they have been described previously but this is the first announcement of boundary coupling etc and so it seems odd.

Perhaps this section needs to be reordered to come after the descriptions.

We have removed these words as they are redundant if come after descriptions. We have added some words in the initial of Section V as follows.

In this section, the procedure to simulate ultrasonic vibration breaking rock are proposed. To

ensure the pre-requisites determining the success of modelling, four key points should be considered.

50. Paragraph 2 Section V. B → Do these conditions also apply to all of the previous models that were presented? Hence, shouldn't this explanation be given earlier?

Although the quasi-static conditions have been applied to the previous models (i.e. UCS, Brazilian and direct tension tests simulations), the purpose of this paragraph was used to introduce one new element (rblock) as ultrasonic vibration loading condition. Thus, we think we need to first explain why we do not use the previous wall element as loading condition. We think it's reasonable to keep it as origin.

51. Paragraph 4 Section V. B → So i'm confused why the two previous paragraphs weren't given in the description of those earlier numerical simulations.

To clear the purpose of these two paragraphs, we have done some revision on them.

The purpose of Section IV is describing the procedure to establish rock model and we should conduct calibration simulations (i.e. UCS and Bralizan test simulations). As these simulations are in static or quasi-static situation, the wall element can be used to simulate the loading condition. However, the ultrasonic vibration is one kind of dynamic condition, thus, we should use another suitable element. These two previous paragraphs introduce the principle to use the new rblock element to simulate the dynamic loading condition.

52. Section V. C → Why is the boundary only applied at the bottom of the model not the top where the loading surface is? Are you not concerned that breakages at the top of the model are due to coupling effects?

Was this boundary purely numerical or did you also use something similar in the lab experiments?

The wave propagates through the underground infinite medium. When using a computer to perform numerical calculations, the calculation domain is limited, so artificial boundaries need to be introduced. The introduction of artificial boundaries will cause strong reflections when waves pass through them, and the waves reflected back to the computational domain. This is unrealistic and will seriously interfere with the results. In order to obtain clear and accurate simulation results, it is necessary to take appropriate measures to deal with these boundaries in order to absorb reflected waves on artificial boundaries. And in our work, we used the viscous boundary to achieve it.

53. Paragraph 1 Section V. E 1) → It is not clear what the authors mean here. If the material is being unloaded then the loading curve is one of reducing stress and hence the stress 'curve' will simply be whatever the unloading 'curve' is. Please clarify.

We have rewritten this sentence as follows. Thanks.

According to Sadd et al.'s research [14], **when rock sample subjects to the cyclic loading**

condition, during one cycle, due to the dissipation of energy, the unloading stress-strain path lies below the loading curve, and damage accumulates.

54. Paragraph 1 Section V. E 2) → What is a 'small range'? Tens, hundred or thousands of cycles? Please be quantitative.

The 'small range of load cycles' here means if we jump with such cycles, the simulation results are not affected. Thus, this range is different in different situations (e.g. with different frequencies and amplitudes). In our case, as the residual strain rate is approximately $10 \mu\epsilon/s$, when we chose 500 as the jumped cycles, the strain would only change $0.16 \mu\epsilon$ per simplified cycle and we assume this simplification is reliable.

55. Paragraph 2 Section V. E 2) → So how many cycles ran in each jump, 500 or 60?

If we do not use jump technique, there are 30000 cycles in each second. When using this technique, there are 60 cycles in each second. Among each jump, there is one cycle and jumps 500 cycles.

56. Paragraph 1 Section VI A → Why so few? In the earlier simulations the number of circles seemed many times greater.

The measurement circles mentioned here were used to obtain strain and stress fields. According to previous research, the minimum size of the measurement circles should include 20–30 particles. The chosen of circle diameter in our work is based on this principle and this would decide the number of circles.

Related reference:

1. Jia, M., Yang, Y., Liu, B. et al. PFC/FLAC coupled simulation of dynamic compaction in granular soils. *Granular Matter* 20, 76 (2018). <https://doi.org/10.1007/s10035-018-0841-y>
2. Kang, C., Chan, D. Numerical simulation of 2D granular flow entrainment using DEM. *Granular Matter* 20, 13 (2018). <https://doi.org/10.1007/s10035-017-0782-x>

57. Paragraph 1 Section VI A → Are you not concerned that since most of the damage occurs at the sample edges, these are simply edge effects related to the loading regime? How can that problem be diagnosed?

The samples were carefully cut and polished according to ISRM(UCS) standard. Thus, the edge effect should not be so much at this situation.

58. Paragraph 1 Section VI B → Were AE's measured in the lab experiments? This is the first mention of AE's it seems.

As this project is the first one related to rock mechanic in our research group, we do not have AE apparatus now. However, it's our follow-up plan as we think this may be an effective method to monitor the crack propagation. Thanks.

59. Paragraph 2 Section VI B → What assumption are you making about the relation between AE output and crack generation? Do you assume that each AE event is a crack or is there some ratio of number of AE's to cracks? Also, do you consider the size of AE events and their relation to different crack growth increments?

Lockner (1993) suggested that only about 1% of AE corresponds to crack growth. This is discussed in detail in Browning et al (2017).

Lockner, D., 1993, December. The role of acoustic emission in the study of rock fracture. In International Journal of Rock Mechanics and Mining Sciences & Geomechanics Abstracts (Vol. 30, No. 7, pp. 883-899). Pergamon.

Browning, J., Meredith, P.G., Stuart, C.E., Healy, D., Harland, S. and Mitchell, T.M., 2017. Acoustic characterization of crack damage evolution in sandstone deformed under conventional and true triaxial loading. Journal of Geophysical Research: Solid Earth, 122(6), pp.4395-4412.

In our simulations, the AE count is defined as the new generated crack number in each 50 timestep, thus, it can be seen as the crack generation rate. The real simulation of AE method is still a difficulty for us, thus, we have no idea about the AE events and their relation to different crack growth increments. We would further study to explore this problem.

We have clarified the AE count again in our manuscript. Thanks.

60. Paragraph 2 Section VI B → It's not clear what this 'local damage' means. What is meant by 'introduced', is this related to the AE somehow? This is very unclear.

The local damage is used to reflect the damage level in different parts of rock sample. We have rewritten this sentence as follows.

To further detail the development of cracks in each stage during the UVRB process, **the damage in different position of rock sample (local damage) was also monitored.**

61. Paragraph 3 Section VI B → Please be quantitative.

Thank you for your suggestion. The revision is as follows.

Accelerating crack stage. It can be seen from the AE frequency that the microcracks are induced at a rate of **11.8/s.**

62. Paragraph 3 Section VI B → Do you mean that they propagate horizontally, i.e. with their a axis growing in the x plane?

Or

Do you simply mean that the right side of the sample contains more fractures?

Please clarify.

The revised content is shown as follows.

It can be seen from the AE frequency that the microcracks are generated at a high rate. The orientations of the tensile cracks are the same as those of the previous stage, **that is parallel and sub-parallel to the loading direction**. Additionally, the cracks gradually propagate **from top-left area** to the right area.

One mark was added to the Figure 20a to improve its readability.

63. Paragraph 3 Section VI B → Please be quantitative.

Thank you for your suggestion. The revision is as follows.

Stable stage. Influenced by the effective region, the cracks are generated slowly **and at a rate of 7/s**.

64. Paragraph 3 Section VI B → Please rephrase.

The revision is as follows.

The upper part of the sample is **almost covered by the cracks**.

65. Paragraph 3 Section VI B → What do you mean by 'effective'?

Inside this effective region, there are obvious damage, while there is almost no damage in other place. We have clarified it the first time it appears. Thanks.

66. Paragraph 4 Section VI B → What does it mean for the strain limitation to decrease? Presumably the limitation is fixed, but perhaps it is unclear what you mean by 'strain limitation'. Please clarify.

The revision is as follows.

As the loading continues, the tensile strain increases, and the tensile strain limitation, which **corresponding to the contact bond strength** between particles [54], decreases **because of the generation of fatigue damage of rock materials under ultrasonic vibration loading**.

67. Paragraph 1 Section VII → What is meant by 'violent' in this context? That wasn't described earlier.

We have replaced this word with 'obvious'. Thanks. The revision is as follows.

Tested intact granite is a typical brittle rock that undergoes **obvious** local failure (**upper part of the sample**) under ultrasonic vibration.

68. Paragraph 1 Section VII → Are you talking about the numerical simulations, the experiments or both here? Which strain curves?

It's the experimental one and we have clarified in the manuscript. Thanks.

69. Paragraph 1 Section VII → Wasn't the compressive stress caused by the compressive loading?

The compressive deformation was due to the non-linear property of rock material. When subject to ultrasonic vibration, some energy is consumed in the unloading process in each cycle and irreversible compressive deformation gradually accumulates and increases.

70. Paragraph 1 Section VII → This is very confusing. Even though there are apparently locally areas of extension within the rock, whilst the material is being compressed the overall deformation is compression. That would be confirmed if a strain gauge was also placed in the central part of the sample, not just in the upper section.

The reason why we placed strain gauge in the upper section is due to the quick attenuation property of ultrasonic vibration, and this has been verified by the strain amplitude test in different height of sample. Besides, as we have cut and polished samples carefully based on ISRM (UCS) standard, the edge effect should not be so much. Thus, the phenomena that the compressive deformation decrease is results from the initiation and propagation of cracks.

71. Paragraph 1 Section VII → Which point? Please be specific.

Due to the difference of lithology, the place of the point is with uncertain property. We have rewritten this sentence as follows.

Finally, with the further of crack initiation, propagation and coalescence, large fracture forms at the edge and then propagate to the adjacent area.

72. Paragraph 1 Section VII → Rephrase.

The revision is as follows.

Thus, the rock fatigue will accelerate due to the increase in the stress amplitude, and this will further accelerate the rock breaking process.

73. Figure 5 → What do the red and blue lines on this plot indicate?

The red and blue lines are the slope of the strain curve to reflect the change of the strain curve. According to your next suggestion, we have deleted these two lines. Thanks.

74. Figure 5 → It would be better if all of these plots had the same range on the y and x axis axes so that they can be compared. For example, the amount of strain experienced in test No.1 is substantially greater than that in test No. 6 even though test No. 6 ran for approximately double the time of test No. 1.

From the visualisations that is difficult to discern, as they essentially look the same, but obviously the strain rate is substantially different. This makes me question the repeatability of such tests. Why do you think they are so different?

Thank you for your suggestion, we have put these figures into one.

The rocks are typical heterogeneous and anisotropic materials, which will certainly make the testing results unreproducible. The other factors, e.g. the preparation of samples, the experimental set-up, and environmental effects such as temperature and humidity will also contribute to the discreteness of the testing results. This needs a plenty of tests in a series of future studies, where the samples made from other brittle and relatively homogeneous materials, such as concrete and gypsum, will be adopted.

Besides from this, it can be found from the figure, although there are some differences, the maximum compressive strains of these six samples are almost same. This is consistent with previous research [7], and they concluded that the axial residual strain could be a better option to describe the fatigue behavior of rock than the loading cycle number (i.e. loading time).

Related reference:

Z. Wang, S. Li, L. Qiao and J. Zhao, "Fatigue Behavior of Granite Subjected to Cyclic Loading Under Triaxial Compression Condition," *Rock Mech. Rock Eng.*, vol. 46, no. 6, pp. 1603-1615, 2013, doi: 10.1007/s00603-013-0387-6.

75. Figure 10 → Why are there cracks outside of the sample, in the numerical simulation?

The yellow background is the sample which is contains of numerous particles. In order to make the crack pattern clear, we have not shown the outline of the particles. We have added the illustration in the figure caption. Thanks.

76. Figure 12 → How was this viscous boundary made in the experiments or it is purely numerical? I can't find this described in the manuscript text.

Why is there only a coupling boundary at the bottom of the sample not the top?

The wave propagates through the underground infinite medium. When using a computer to perform numerical calculations, the calculation domain is limited, so artificial boundaries need to be introduced. The introduction of artificial boundaries will cause strong reflections when waves pass through them, and the waves reflected back to the computational domain. It is unrealistic and will seriously interfere with the results. In order to obtain clear and accurate simulation results, it is necessary to take appropriate measures to deal with these boundaries in order to absorb reflected waves on artificial boundaries. And in our work, we used the viscous boundary to achieve it.

77. Figure 12 → This figure is not especially informative. I suggest this can simply be explained in the manuscript text without the need for a figure.

Thank you for your suggestion and we have removed it.

78. Figure 12 → Which experiment are these 'experimental results' from? How did you decide which test to pick as the amount and rate of strain is substantially different between the 6 tests shown.

As explained above, there are numerous factors would influence the discrete of test data for rock materials. However, obvious law still can be found from these experimental data. Based on comparing the strain curve and fragmentation pattern, it can be concluded that the established numerical model could well reproduce the fragmentation and fatigue behavior of brittle rock under ultrasonic vibration. Thus, we can further analyze the fragmentation mechanisms according to this model.

79. Figure 12 → Is this from the experimental or numerical data? If experimental, which test? And why not show the AE output for all the tests?

Can you also plot the AE as 1) a hit rate and 2) cumulative output. I think both are essential to compare to the Damage index.

This is the numerical data of the establish model.

As mentioned above, we do not have AE apparatus now. However, it's our follow-up plan as we think this may be an effective method to monitor the crack propagation. Thanks.

80. Table 2 → Why not use a different symbol for each, to avoid confusion?

Thank you for your suggestion and we have revised to use different symbols.

TABLE II
GRANITE PROPERTIES FROM EXPERIMENTS AND SIMULATIONS

Property	Experiment results	Simulation results
Young's modulus E (GPa)	$52.3 \pm 12.2(m=10)$	$52.7 \pm 0.26(n=10)$
Poisson's ratio m	$0.26 \pm 0.05(m=10)$	$0.255 \pm 0.006(n=10)$
Uniaxial compressive strength (UCS) (MPa)	$116 \pm 19(m=10)$	$116.3 \pm 3.1(n=10)$
Brazilian tensile strength (BTS) (MPa)	$8.1 \pm 1.5(m=10)$	$8.1 \pm 0.17(n=10)$

The m denotes the number of samples in the UCS and Brazilian tests; the n denotes the number of samples with different bond strength distributions in the simulations.

Other changes:

1. The title of Section V. A has been removed and the other in the section V have been reordered.
2. The title of Section VII has been revised as there is one mistake.
3. The serial number of the figures was re-edited because of the remove of two figures.
4. The number of reference 18 has been added in the manuscript.
5. Added some important references:
 - a) Bagde, M.N. and V. Petroš, Fatigue and dynamic energy behaviour of rock subjected to cyclical loading. *International Journal of Rock Mechanics and Mining Sciences*, 2009. 46(1): p. 200-209.
 - b) Wang, H., et al., Effect of the intermediate principal stress on 3-D crack growth. *Engineering Fracture Mechanics*, 2018. 204: p. 404-420.
 - c) Song, H., et al., Damage evolution study of sandstone by cyclic uniaxial test and digital image correlation. *Tectonophysics*, 2013. 608: p. 1343-1348.
 - d) Wang, H., et al., Experimental and Numerical Study into 3D Crack Growth from a Spherical Pore in Biaxial Compression. *Rock Mechanics and Rock Engineering*, 2019.
6. Removed some references with low relation:
 - a) Investigation on Microstructure and Damage of Sandstone Under Cyclic Dynamic Impact
 - b) Numerical Study of Rock-Soil Aggregate by Discrete Element Modeling